# Probing Human Visual Robustness with Neurally-Guided Deep Convolutional Neural Networks

## Abstract

Humans effortlessly navigate the visual world, yet deep convolutional neural networks (DCNNs), despite excelling at many visual tasks, are surprisingly vulnerable to adversarial perturbation that are almost imperceptible and easily handled by humans. Past neuroscience work proposes that the ventral visual stream (VVS) builds an increasingly robust object representation across successive stages. However, this hierarchical refinement across VVS has not been systematically demonstrated in humans, and it remains possible that robustness arises only in specialized higher-order regions such as the inferior temporal cortex (IT). To distinguish these possibilities, we trained DCNNs to align their representations with human neural responses from consecutive VVS regions during visual tasks. We demonstrate a hierarchical improvement in DCNN robustness: alignment to higher-order VVS regions yields greater gains. To investigate the mechanism behind this improvement, we test a prominent hypothesis that attributes human visual robustness to the unique geometry of neural category manifolds in the VVS. We show that desirable manifold properties, specifically, smaller extent and better linear separability, emerge across the human VVS. These properties are inherited by DCNNs via neural guidance and can predict their subsequent robustness gains. Further, we show that supervision from neural manifolds alone, via manifold guidance, suffices to qualitatively reproduce the hierarchical robustness improvements. Together, our results highlight the evolving VVS representational space as critical for robust visual inference, with the more linearly separable category manifolds as one potential mechanism, offering insights for building more resilient AI systems.

## 1 Introduction

Deep convolutional neural networks (DCNNs), despite achieving human-level performance on many visual tasks (Deng et al., 2009; Xu et al., 2015), remain brittle under minor image perturbations that are imperceptible and innocuous to humans, such as adversarial attacks (Papernot et al., 2018; Szegedy et al., 2013; Goodfellow et al., 2014). In contrast, humans are known for their remarkable proficiency in navigating the challenging visual world (Biederman & Cooper, 1991; 1992; Biederman & Gerhardstein, 1993), even in face of evolutionarily unnatural image degradations (Geirhos et al., 2018; Malik et al., 2023; Crowder & Malik, 2022) and novel objects that fall outside of typical visual experience (Bowers et al., 2016; Vuilleumier et al., 2002). This stark contrast is thought to stem from humans' ability to develop a unique representational space that supports robust visual inference. DCNNs, on the other hand, could potentially lack either the necessary architectural components (such as the extensive feedback and recurrent connections (Felleman & Van Essen, 1991)) or the rich training environment (such as the active interaction with real-world (Snow & Culham, 2021)) to naturally achieve such a superior representational space.

Computational models (Serre et al., 2007) and theoretical frameworks (DiCarlo & Cox, 2007; Iordan et al., 2015; Kravitz et al., 2013) have highlighted the ventral visual stream (VVS), comprising hierarchically organized visual cortices such as V1, V2, V4, and the inferotemporal (IT) cortex, as critical for object recognition. Neuroscience studies have shown that representations in higher-order VVS areas become progressively more abstract and invariant to transformations (Rust & DiCarlo,

2010; Zoccolan et al., 2007; Isik et al., 2014; Hong et al., 2016; Bao et al., 2020; Quiroga et al., 2005). However, these previous studies have typically focused on single regions or limited comparisons between two areas (e.g. V4 vs. IT) (Rust & DiCarlo, 2010; Hong et al., 2016), leaving unanswered the fundamental question of **whether** robust visual representations are indeed progressively built along the VVS hierarchy, and if so, **how** this progression occurs. An alternative hypothesis is that robustness emerges exclusively in specialized regions such as IT (analogous to how face processing is localized to fusiform gyrus (Kanwisher et al., 1997)), without contributions from early or intermediate regions. Resolving this question is important for both providing critical mechanistic insights into how human vision achieves resilience, and informing AI research by revealing the plausibility of intermediate representational stages scaffolding robustness in artificial systems.

To systematically evaluate whether robustness emerges progressively across the VVS, instead of solely in a specialized endpoint like IT, we build on prior work using neural alignment to train DCNNs with primate brain data (Li et al., 2019; Safarani et al., 2021; Dapello et al., 2022; Sucholutsky et al., 2023). Specifically, we apply "neural guidance" to steer the final-layer representations of DCNNs to match human neural responses, recorded by functional Magnetic Resonance Imaging (fMRI), from consecutive regions of human VVS. By training a separate model aligned to each region, we are able to systematically test the hypothesis that later brain areas confer greater robustness to adversarial perturbations, thus supporting a VVS hierarchy of neural representations with increasing robustness-supporting capacity.

Moreover, we further probe the mechanism to ask what representational properties may be responsible for the improved robustness. One prominent candidate explanation is the manifold disentanglement hypothesis (DiCarlo & Cox, 2007; Chung et al., 2018; Cohen et al., 2020). In this framework, identity-preserving transformations of an object form continuous manifolds in neural representational space, and robustness emerges as these manifolds become less diffuse, lower-dimensional, and more linearly separable across the VVS hierarchy. While theoretically appealing, direct evidence linking human neural manifolds to robustness has been limited. Here, we first test this connection using mean-field theoretic manifold analysis (MFTMA) (Chung et al., 2018), a recently developed framework that uses statistical mechanical modeling to probe the geometric and statistical properties of manifolds. We measure the extent (i.e., spatial spread) and linear separability of human neural category manifolds across VVS and ask whether such properties can be inherited by DCNNs to predict model robustness. We then introduce "manifold guidance", a form of category manifold-level neural alignment that moves beyond point-to-point neural guidance to test whether the structure of neural manifolds is indeed one critical component underlying robustness. Importantly, our experiments focus on adversarial robustness rather than the broader space of out-of-distribution (OOD) robustness as it remains unclear whether and which OOD transformations remain identity-preserving and thus remain on-manifold, and we provide the reasoning in Appendix F.

Our contributions are as follows:

- We show that DCNNs trained with neural guidance from later VVS regions show progressively greater robustness to adversarial attacks, corroborating a functional hierarchy across human VVS to support robustness.

- We demonstrate that human neural manifolds become less diffuse and more linearly separable across the VVS, and that these properties can be inherited by neurally-guided DCNNs and predict their subsequent robustness gains.

- We introduce "manifold guidance" to show that supervision from coarse neural manifold structure is sufficient to qualitatively reproduce the hierarchical robustness effect, supporting neural manifold geometry as a key factor in human visual robustness.

## 2 RELATED WORK

Neural alignment has been increasingly explored as a method to transfer biological properties to DCNNs (Sucholutsky et al., 2023). Studies have aligned models to neural responses to produce more human-like behavior, such as object recognition decision patterns (Dapello et al., 2022; Lu & Wang, 2025), shape-over-texture preferences (Khosla et al., 2022), and even semantic biases in language models (Moussa et al., 2025). Work closer to robustness has used second-order similarity in representational geometry (Kriegeskorte et al., 2008) to regularize DCNNs (Li et al., 2019), targeted

early DCNN layers (Safarani et al., 2021) or leveraged specialized architectures such as CORnet whose layers have theoretical correspondence to specific brain areas (Dapello et al., 2022), showing improved robustness.

However, most such studies aim to demonstrate that neural alignment can induce primate-like downstream behavior, by focus on a single brain area, such as V1 or IT, and rely on animal electrophysiology. In contrast, we leverage human fMRI data that allow simultaneous investigation of multiple fine-grained ventral stream regions, enabling a systematic test of whether robustness increases with alignment to progressively higher VVS areas. Beyond demonstrating this effect, we further investigate its underlying mechanism, i.e., whether it arises from the geometry of neural category manifolds.

Regarding biological manifolds, Froudarakis et al. (Froudarakis et al., 2020) observed that higher-order visual areas in rodents show less diffuse manifolds than earlier ones. Here, we extend this to human VVS using MFTMA, demonstrating the decreasing extent and increasing linear separability of neural manifolds. Prior work (Dapello et al., 2021) applied MFTMA to DCNNs robustified with adversarial training and found improved manifold statistics, including more compact manifolds and higher linear separability compared with vanilla DCNNs. These results suggested a possible connection between manifold structure and robustness. However, this work did not test whether neural manifolds across the human ventral visual stream exhibit similar properties, nor whether explicitly imposing such manifold properties alone is sufficient to confer robustness. We extend these efforts by introducing manifold guidance to directly test whether neural manifold geometry alone is sufficient to induce robustness. Additionally, while MFTMA has inspired self-supervised learning methods (Yerxa et al., 2023; Schaeffer et al., 2024), we use it here as an empirical tool to evaluate whether the robustness-supporting capacity of the human brain can indeed be explained by this framework.

## 3 METHODS

### 3.1 NEURAL GUIDANCE

**Neural data** We used neural data from the publicly available Natural Scenes Dataset (NSD) (Allen et al., 2022), which provides high-resolution, whole-brain fMRI scans of human participants viewing natural images from MSCOCO (Lin et al., 2014). In NSD experiments, participants viewed a single image in each trial, and voxel-wise activation levels were estimated via a general linear model (GLM) with a hemodynamic response function regressor and nuisance regressors for denoising. Activation levels correspond to the GLM coefficients for each image and reflect the percent change in fMRI signal magnitude relative to a voxel baseline activity. For each brain region, or Region of Interest (ROI), these values across voxels form a neural representation vector per image. We used the fully preprocessed data and signal estimates provided by the NSD pipeline to ensure reproducibility. To trace evolving representational spaces across the VVS, we extracted neural responses from seven visual ROIs (Fig. 1A), ranging from early visual areas (V1, V2) to intermediate region (V4), higher-order areas (VO, PHC, LO, TO (see Appendix A for detailed descriptions and expected contributions to robustness). All experiments reported here are based on Subject-1

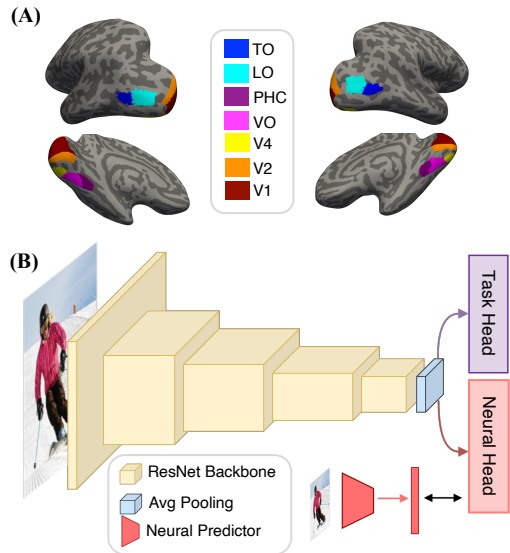

Figure 1: **(A)** Illustration of anatomical locations of seven ROIs from Subject-1 used for neural guidance. **(B)** Architecture of the neurally guided ResNet, trained with a task head for image classification and a neural head to match neural responses generated by neural predictors.

as optimal registration across individuals remains a challenge (Haxby et al., 2020). However, results are similar for other NSD subjects (Appendix C.1.)

**Neural predictors**    Following prior neural alignment work (Safarani et al., 2021; Dapello et al., 2022), we used the human neural data acquired in NSD to train [1] neural predictors, serving as surrogates for each ROI. These neural predictors enabled us to obtain neural responses for any arbitrary images, such as those from the ImageNet dataset, that are not seen by human participants but required for the neural guidance training and subsequent evaluation on DCNN robustness. Following the NSD experiments' protocol for train/test split, we divided the 9,841 unique images viewed by Subject-1 into a training set with 8,859 image-representation pairs and a test set with 982 pairs. Separate predictors were trained for each ROI (seven in total) based on a modified ResNet18 architecture (He et al., 2016), in which the final fully connected (FC) layer was replaced with a linear layer matching the targeted ROI's dimensionality (e.g., the V1 predictor output matches the dimensionality of V1 representations). All classification-specific components (e.g., softmax) were removed. Each predictor was trained to minimize a Mean Squared Error (MSE) loss between the predicted and ground-truth neural representations. After training, predictor weights were frozen and used as feature extractors for subsequent neural guidance training. We performed quality control analyses to confirm that predictors indeed captured meaningful neural representations (Appendix B).

**Neural guidance (NG)**    As shown in Fig. 1B, we adapted ResNet-18 by adding an additional FC layer, the "neural head", situated after the last convolutional block to output representations matching the dimensionality of the targeted ROI, while the original "task head" performs classification. Training thus jointly optimized the classification and neural representation alignment objectives. Specifically, given an input $x_i$, its class label $y_i$ and neural representation $r_i$, the shared backbone $f^s$ produces features passed to both the task head $f^t$ and the neural head $f^n$. The overall loss is:

$$L = \alpha L_{\text{CE}}(f^t(f^s(x_i)), y_i) + (1 - \alpha)L_{\text{MSE}}(f^n(f^s(x_i)), r_i), \tag{1}$$

where $L_{\text{CE}}$ is the cross-entropy loss for classification and $L_{\text{MSE}}$ denotes the MSE loss for representation matching. $\alpha$ was tuned to balance the two objectives.

We trained seven NG-models, each guided by one ROI along the VVS hierarchy to test the effect of neural guidance from different stages of the ventral stream. All models were trained on a curated ImageNet subset (ImageNet-50) selected to match the object categories used in NSD. Since MSCOCO images used in NSD do not map directly to ImageNet, we classified them using a ResNet-101 trained on ImageNet and selected the 50 most frequent categories. This procedure aimed to maximize the generalizability of neural predictors from MSCOCO to ImageNet, thus improving the quality of approximated neural responses. In addition to ImageNet-50 classification, we applied the same neural guidance setup to CIFAR-100 classification and MSCOCO image captioning (evaluated by BLEU scores (Papineni et al., 2002)), with results reported in Appendix C.3.

We also included four baseline models to rule out alternative hypotheses. The "None" model, serving as the absolute baseline, had only the task head and was merely fine-tuned on the ImageNet-50 subset. The "Random" model had the same dual-head structure, but is neural head was guided by an untrained neural predictor with randomly initialized weights to test the impact of an uninformative auxiliary head. The "V1-shuffle" and "TO-shuffle" were both dual-headed models guided by frozen V1 and TO neural predictors, respectively. However, before training, we shuffled the correspondence between input images and neural representations to pair each image, consistently, with an incorrect target. This tested whether improvements could arise from the neural-characteristic noise.

**Robustness evaluation**    During robustness evaluation, the neural head was removed from all NG models, and adversarial attacks were applied solely to the task head. Our primary attack was the untargeted $l_\infty$-based Projected Gradient Descent (PGD) attack (Madry et al., 2017). We swept across a range of perturbation bounds $\epsilon$ ($\epsilon = 0.001*(2i+1)$, where $i \in \{0, \ldots, 9\}$) for more comprehensive evaluation. To confirm robustness gains were not specific to PGD, we additionally included several common adversarial attacks : 1. $l_\infty$-based AutoAttack (Croce & Hein, 2020), which aggregates FAB, Square Attack, APGD-CE, and APGD-DLR; 2. $l_\infty$-based Fast Gradient Sign Method (FGSM) (Goodfellow et al., 2014); 3. $l_2$-based Fast Gradient Method (FGM) attack (Goodfellow

---

[1]All models in our study were trained using 4 NVIDIA A40 GPUs.

et al., 2014); and 4. $l_2$-based DeepFool attack (Moosavi-Dezfooli et al., 2016). Robustness gains under all attacks are reported in Appendix C.2.

### 3.2 Neural category manifold disentangling hypothesis

**Manifold statistics**    To test the manifold disentanglement hypothesis (DiCarlo & Cox, 2007), we first applied MFTMA (Chung et al., 2018) to assess whether category manifolds with smaller extent and greater linear separability indeed emerge in both human VVS ROIs and neurally guided DCNNs. A category manifold is defined as the continuous surface comprising representations of all identity-preserving transformations of objects within a category. MFTMA estimates the system's capacity, or linear separability, that reflects the maximal number of manifolds that can be linearly separated in an $N$-dimensional space, based on manifold geometry. Specifically, MFTMA estimates each manifold's radius $r_c$, which reflects the size of the manifold for category $c$, effective dimension $d_c$ for the number of dimensions with significant variance, and thus extent $e_c = \frac{r_c}{\sqrt{d_c}}$ that captures the spatial diffuseness of the manifold. In particular, a more optimally structured representational space should have more compact manifolds (smaller $e_c$) to support better linear separability.

We first applied this analysis to human voxel activity patterns in each of the seven ROIs using NSD. Because most MSCOCO images in NSD contain multiple objects and thus are not ideal samples for capturing object category manifolds, we curated a subset of images containing only a single object category (see Appendix E.1), by prioritizing those with large bounding boxes. This resulted in 12 object categories, each with 50 images. We also applied MFTMA on neural representations estimated by the neural predictors using the same subset of images to confirm that they preserve the structure of the original neural manifolds.

To assess manifolds in the NG-models, we used the test set from the ImageNet-50 subset, which provided 50 categories and 50 images per category. To further test the manifold hypothesis (Chung et al., 2018; Dapello et al., 2021) that identity-preserving transformations should be treated as part of the same manifold, we enriched each category with adversarially perturbed variants of the test images, thus obtaining 100 samples per category, similar to previous work (Dapello et al., 2021). These perturbed versions of test set were generated using the $l_\infty$-based PGD attack with $\epsilon = 0.009$.

**Manifold guidance**    To test whether neural category manifold structure is sufficient to confer robustness, we replaced the neural guidance loss in Eq. 1 with constraints that match key geometric properties of DCNN category manifolds to those of the corresponding neural ones (see Fig. 4A). Importantly, we approximate each manifold as a first-order linear structure, defined by its center, spread (radius), and local subspace. This simplification allows for efficient integration into DCNN training and is sufficient, as a proof-of-concept, to assess whether coarse geometric structure alone can induce robustness improvements.

We first characterized each neural category manifold $c$ offline using neural representations of the training set of ImageNet-50 images (i.e., 50 categories, 1300 clean image samples per category). For each category, we estimated 1. the center $m_c$ as the geometric mean of all samples, 2. the radius $r_c$ as the root-mean-squared (RMS) spread of centered representation vectors and 3. the subspace basis $V_c$ as the top singular vectors from singular value decomposition (SVD) retaining enough components to explaining 99% of variance.

During training, for each batch of images $X_b$ with labels $Y_b$, we minimize the loss:

$$L_b = \alpha L_{\text{CE}}(f^t(f^s(X_b)), Y_b) + (1 - \alpha) \sum_{c \in \text{set}(Y_b)} L_{\text{manifold}}(f^n(f^s(X_{b,c})), m_c, r_c, V_c), \quad (2)$$

where $\text{set}(Y_b)$ denotes the unique object categories present in batch $b$, which, given a sufficiently large batch size, typically includes all 50 categories in the dataset, and $L_{\text{manifold}}$ enforces alignment between DCNN category manifolds and neural manifolds. In a given batch, we grouped images based on their category $X_{b,c}$ and treat them as samples for the DCNN manifold of category $c$. The manifold guidance loss were computed separately for each category and consisted of two terms:

1. a radius constraint that penalizes if the DCNN manifold radius exceeds the neural one

$$L_{\text{RMS}}(Z_{b,c}, m_c, r_c) = \max(0, f_{\text{RMS}}(Z_{b,c} - m_c) - r_c), \quad (3)$$

where $Z_{b,c} = f^n(f^s(X_{b,c}))$ and $f_{\text{RMS}}$ denotes the root-mean-squared error,

2. a subspace constraint, analogous to the Variance Accounted For (VAF) (Elsayed et al., 2016; Gallego et al., 2018), that encourages orientation alignment between the DCNN and neural manifolds:

$$L_{\text{subspace}}(Z_{b,c}, V_c) = \frac{\|Z_{b,c} - V_c V_c^\top Z_{b,c}\|_F^2}{\|Z_{b,c}\|_F^2}, \tag{4}$$

where $\|\cdot\|_F$ denotes Frobenius norm.

The total manifold guidance loss is then summed over all categories in the batch

$$L_{\text{manifold}}(Z_{b,c}, m_c, r_c, V_c) = L_{\text{RMS}}(Z_{b,c}, m_c, r_c) + \beta L_{\text{subspace}}(Z_{b,c}, V_c), \tag{5}$$

where $\beta$ was tuned to balance the weighting between radius and subspace constraints. We note that to reliably estimate DCNN category manifolds in each batch, the batch size needs to be large enough to allow a reasonable sample size for each category.

## 4 RESULTS

### 4.1 NEURAL GUIDANCE FROM HUMAN VVS INDUCES HIERARCHICAL IMPROVEMENT IN DCNN ROBUSTNESS

We first tested whether direct neural guidance from later regions of the VVS confers greater adversarial robustness to DCNNs. As described above, we trained seven neurally guided DCNNs, each using a frozen neural predictor serving as a surrogate for a different ROI along the VVS (V1–TO), while simultaneously learning image classification on ImageNet-50. We additionally included four control models, None, Random, V1-shuffle and TO-shuffle to rule out various hypothesis.

Fig. 2A shows performance of all models under the $l_\infty$-based PGD attack on the classification head with varying perturbation strengths $\epsilon$. While all baseline models showed similar vulnerability, all NG-models exhibited improved robustness. More importantly, this improvement followed a clear hierarchy. DCNNs guided by V1 and V2 showed modest improvement in adversarial robustness, consistent with their early positions in the visual processing hierarchy. Improvements were more pronounced in the V4-guided model and were further surpassed by the LO-guided, VO-guided, and PHC-guided models, with the TO-guided model showing the highest level of robustness. Notably, we also observed a hierarchical, although modest, decrease in clean accuracy for these neurally guided models. This pattern is consistent with the well-known tradeoff between clean and adversarial accuracy (Tsipras et al., 2018; Zhang et al., 2019; Ilyas et al., 2019), as shifting models towards more stable, human-aligned features reduces reliance on non-robust but highly predictive cues that may inflate clean accuracy.

We replicated this hierarchical effect across multiple NSD subjects (Appendix C.1), alternative tasks and datasets (Appendix C.3), diverse attack types (Appendix C.2), across different random seeds (Appendix C.4) and on a different DCNN architecture, VGG-16 (Simonyan & Zisserman (2014), Appendix C.5). In all cases, robustness increased with guidance from later visual areas, despite the absolute gains varying across settings. Furthermore, we show in Appendix B that such hierarchical robustness gains could not be explained by the neural predictor performance or the fMRI signal quality in each VVS region. These converging findings show that the robustness conferred by neural guidance is not coincidental. Instead, they suggest that human-constructed visual representations along the VVS become increasingly capable of supporting visual robustness and that such property is transferable to artificial systems.

### 4.2 NEURAL GUIDANCE INDUCES UNIQUE OUTPUT SURFACES AND REPRESENTATIONAL SPACES

One possible explanation for the robustness improvement observed in NG-models is that they stem from standard regularization effects. For example, simply increasing weight decay (WD) is a common approach known to improve adversarial robustness by smoothing the model's output surface, i.e. loss surface with respect to input images (Rosca et al., 2020; Loshchilov & Hutter, 2017; Cohen et al., 2019; Yu et al., 2019). To assess whether neural guidance offers a distinct solution to robustness, we compared the output surface characteristics of NG-models and DCNNs trained with increasing WD regularization.

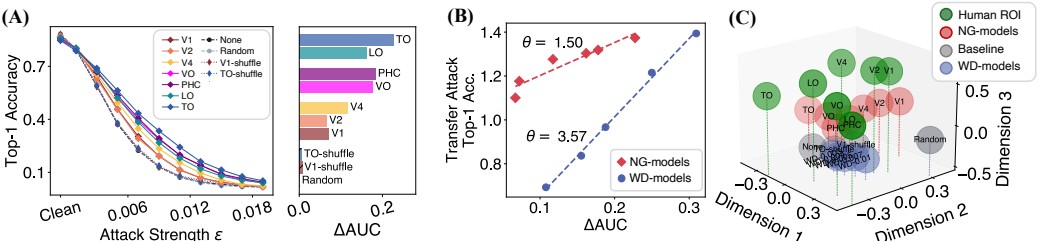

Figure 2: **(A)** Left: Top-1 classification accuracy of all models under $l_\infty$-based PGD attacks of varying strength $\epsilon$. For reference on the perceptibility of attack magnitudes, see Appendix G. "Clean" denotes accuracy on unperturbed images. Right: Robustness improvements summarized as the difference in area under the accuracy curve ($\Delta$AUC) for each model relative to the baseline (None model), computed across all $\epsilon$ values used (see Methods 3.1). **(B)** Transfer attack accuracy vs. native robustness improvement ($\Delta$AUC) for NG- (red diamonds) and WD-models (blue circles). Dashed lines show linear fits with slopes $\theta$ annotated (see Results 4.2). **(C)** 3D MDS visualization of representational space similarity among 16 models and 7 human ROIs. Proximity indicates similarity in the representational space. Dashed stems are added for visualization of the relative positions of each circle.

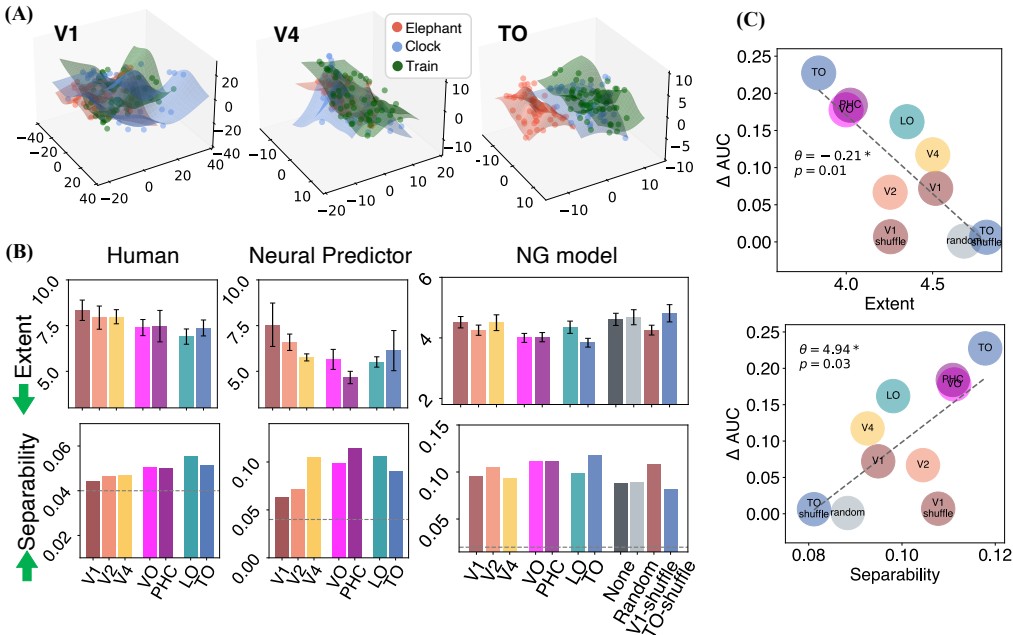

Figure 3: **(A)** Isomap visualization of three example category manifolds in V1, V4, and TO representational spaces (note the shrinkage in axis scales indicating smaller extent) **(B)** Category manifold extent ($\downarrow$) and linear separability ($\uparrow$) estimated using MFTMA (Chung et al., 2018) for human brain ROIs (left), neural predictors (middle), and neurally-guided (NG) models (right). Error bars represent 95% Confidence Interval (CI) of extent across categories. Dashed lines mark the chance level for separability. **(C)** Manifold extent and linear separability in NG models predict their robustness improvement ($\Delta$AUC): NG models with less diffuse (smaller extent) and more separable manifolds show greater $\Delta$AUC.

We first trained five WD models with increasing regularization strength and confirmed that stronger regularization improved adversarial robustness (see Appendix D.1). We also verified that both WD and NG models showed smoother output surfaces, , measured by gradient norm magnitude across neighborhood radii (Yang et al., 2021) (see Appendix D.2). To probe whether the output surface geometry differs between NG and WD models, we leveraged transfer attacks (Papernot et al., 2016), which test whether adversarial examples generated for one model can also compromise others. Transferability of adversarial examples is a hallmark of conventional DCNNs (Papernot et al., 2016; Lu et al., 2020; Richards et al., 2021), suggesting a fundamental homogeneity among them. We generated adversarial examples from the baseline None model and evaluated them on NG and WD models (see Appendix D.3 for details). Importantly, we matched models based on their native robustness improvement ($\Delta$ AUC, the difference in area under the PGD accuracy curve relative to the None model) to isolate transfer susceptibility. We also fit linear regressions between $\Delta$AUC and transfer attack accuracy for NG and WD models. As shown in Fig. 2B, NG models were consistently more resistant to transfer attacks than equally robust WD models and had flatter regression slopes ($\theta = 1.50$) compared to WD models ($\theta = 3.57$), indicating that their transfer susceptibility scales less with native robustness. Together, these results suggest that neural guidance alters the output surface in a qualitatively different way compared to common regularization effects.

We hypothesized that these output surfaces differences arise from changes in representational geometry induced by neural guidance. Using representational similarity analysis (RSA) (Kriegeskorte et al., 2008) (methods in Appendix D.4), we compared internal feature spaces across 16 models (NG, baseline, WD) and 7 human ROIs, deriving a Representational Similarity Matrix (RSM) using Spearman's $\rho$ correlation. We then applied multidimensional scaling (MDS) to visualize the similarity structure in 3D space (Fig. 2C). Our results showed that WD and baseline models clustered closely, whereas NG models diverged and shifted toward human ROIs. This shift suggests that neural guidance steers DCNNs towards more human-like internal representations, potentially producing the distinct loss landscapes that underlie their robustness.

Notably, there exist more powerful and established regularization-based defenses that protect against adversarial attacks, in particular adversarial training (Madry et al., 2017; Ribeiro et al., 2023), which introduces worst-case perturbed images during training. We therefore examined whether neural guidance might behave in a similar way by analyzing the representational space of adversarially trained models. Specifically, we trained three additional models with adversarial training at different perturbation strengths (see Appendix D.5). As expected, adversarial training led to strong robustness under the $l_\infty$ based PGD evaluation (Fig. 15A). However, its representational geometry closely matched that of the WD models and the baseline model (Fig. 15B), without shifting towards the geometry observed in the human VVS or in the neurally guided models. This parallel reinforces the idea that VVS representations could be a distinct solution to robust visual inference that differs in fundamental ways from common engineering-driven defense techniques.

### 4.3 Neural manifold geometry predicts robustness in NG models

Having shown that neural guidance improves robustness and alters representational geometry, we next asked: what specific properties of human neural representations make them particularly effective for supporting robustness? As described above, the manifold disentangling framework (DiCarlo & Cox, 2007; Chung et al., 2018) suggests that category manifolds, encompassing all identity-preserving transformations of objects in the given category, are progressively disentangled across the human VVS, naturally leading to robustness. This disentanglement is thought to be achieved by transforming manifolds to have smaller extent and thus become more linearly separable. To illustrate this visually, we show an example in Fig. 3A using Isomap (Tenenbaum et al., 2000) to embed three category manifolds into 3D space from V1, V4, and TO representational spaces (see Appendix E.2 for details). As we move up the VVS hierarchy, the manifolds appear progressively less diffuse (note the shrinkage in axis scale) and separated.

Here using MFTMA, we quantified two key manifold statistics: extent, which reflects the diffuseness of each category manifold, and linear separability for all manifolds in the representational space. First, we observed in human neural space that manifold extent decreased and separability increased along the VVS (Fig. 3B left), confirming that later brain regions encode increasingly disentangled category manifolds. Similar patterns of decreasing extent and increasing separability in

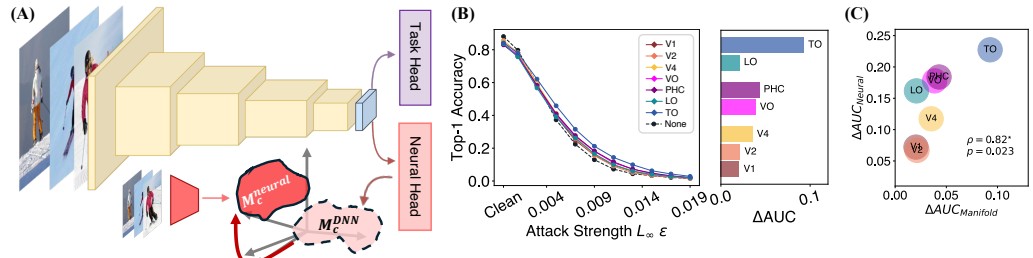

Figure 4: **(A)** Illustration of manifold guidance that trains DCNN with a task head for image classification and a neural head to match category manifold properties with the corresponding neural manifold ones estimated using neural predictors **(B)** Left: Top-1 classification accuracy of models trained with manifold guidance and the baseline None model under $l_\infty$-based PGD attacks. "Clean" denotes accuracy on unperturbed images. Right: Robustness improvements summarized as $\Delta$AUC relative to the baseline None model. **(C)** Correlation of robustness improvement $\Delta$AUC for DCNNs trained with manifold guidance vs. neural guidance. Improvement hierarchy is preserved despite the small improvement magnitude.

neural predictors (Fig. 3B middle) confirms that they retain key geometric features of the original neural space.

Similarly, NG models guided by later brain regions showed lower manifold extent and higher separability than those guided by earlier regions (Fig. 3B right). In comparison, baseline models overall showed diffuse, poorly separated manifolds. To quantitatively test whether these manifold properties explain NG models' robustness gains, we fit linear regressions to predict robustness improvement $\Delta$AUC using manifold extent and separability. As expected, models with less diffuse and more linearly separable manifolds showed greater robustness improvements (Fig. 3C). Together, these results show that the desirable manifold properties observed in the human VVS can be inherited by NG models and can predict their downstream robustness gains.

### 4.4 MANIFOLD-LEVEL GUIDANCE QUALITATIVELY REPRODUCES ROBUSTNESS HIERARCHY

As proposed by the manifold disentangling hypothesis (DiCarlo & Cox, 2007; Chung et al., 2018), the key factor enabling visual robustness is the linear separability of category manifolds. Therefore, the robustness gains observed in neural guidance may arise from learning the more optimal structure of neural category manifolds, such as their smaller extent, orientation, and resulting separability. Importantly, if the geometry of the manifolds is what matters, then matching only manifold-level properties, instead of precisely matching individual representations, should suffice to recover the hierarchical robustness improvement. Motivated by this, we used a coarser form of supervision, manifold guidance Fig. 4A, to align the overall structure of category manifolds between DCNNs and human neural representations.

As shown in Fig. 4B, models trained with manifold guidance showed a hierarchy in robustness: guidance from later visual areas produced larger gains. Although the overall robustness improvements were smaller than those achieved with full neural guidance, the relative ordering across brain regions was well preserved. Robustness gain ($\Delta$AUC) under manifold guidance was highly correlated with that under neural guidance from the same regions (Spearman's $\rho = 0.82^*, p = 0.023$, Fig 4C), indicating that the hierarchical effect from neural guidance is indeed reproduced.

It is important to note that our manifold guidance method uses a linear approximation of manifold geometry and does not capture nuanced nonlinear features such as curvature. Nonetheless, preserving only coarse geometric structure of neural manifolds, such as radius and subspace orientation, without access to any individual neural activation patterns, is already sufficient to recover the hierarchical robustness improvement. This provides strong evidence that learning the neural manifold properties could indeed contribute to the robustness conferred by neural guidance. However, the fact that manifold guidance does not fully reproduce the effects of neural guidance suggests promising directions for future work, such as incorporating more flexible or nonlinear manifold constraints, or investigating the role of finer-grained structures tied to individual neural representations.

## 5 DISCUSSION AND CONCLUSIONS

In our study, we first demonstrated that aligning DCNN representations to progressively higher-order regions of the human VVS yields a hierarchical improvement in robustness. We further showed that this progression is not merely a byproduct of generic regularization, but may be driven by the formation of category manifolds with smaller extent and greater linear separability. These results suggest that robustness may emerge through a progressive refinement of representations along the VVS, underscoring the importance of studying the visual system systematically rather than focusing on isolated or endpoint regions. Our findings also point to the geometry of human VVS representations as a potential source of robustness-supporting principles, thus offering promising neural priors and manifold-level constraints to inspire the development of more resilient AI models.

One limitation of our study is that while we show neural manifold structure relates to visual robustness we do not address how such manifolds form in the brain. Although our findings suggest a feed-forward progression across VVS, top-down influences likely play a key role. The brain has extensive feedback and recurrent connectivity (Felleman & Van Essen, 1991), even reaching early regions like V1 (Lee & Mumford, 2003), which may carry more sophisticated and abstract information. This could explain the robustness gains from V1 despite its early position along the VVS. Indeed, recent work has shown adding feedback improves DCNN performance (Konkle & Alvarez, 2023; Shi et al., 2023). One particularly relevant study demonstrated that feedback in linear neural networks adapts representation manifolds to preserve robustness under noise (Naumann et al., 2022). This also aligns with a long-standing view of the visual brain as a generative system that forms and tests perceptual hypotheses informed by both sensory stimulation and environmental statistical regularities (Beck et al., 2024; Von Helmholtz, 1925). Future work could examine how specific top-down influences, such as prior knowledge or expectations, guide robust manifold formation and how such generative principles might be incorporated in artificial systems.

Another limitation of our approach is that the neural signals used for guidance come from fMRI data rather than direct electrocorticography recordings, which have been used in much of the prior work on non-human primates (Li et al., 2019; Dapello et al., 2022). fMRI measurements are known to have relatively coarse spatial and temporal resolution and the BOLD responses only provide an indirect estimate of neural activity (Logothetis, 2008). Although we take advantage of a high resolution and unprecedentedly large dataset (Allen et al. (2022)), these constraints remain and may contribute to the modest absolute gains produced by neural guidance. However, fMRI offers a unique advantage that is essential for the present study. That is, it provides simultaneous coverage of multiple ventral stream regions in the same human participants at a fine enough spatial scale to reveal the representational progression across the hierarchy, which is central to our goal of establishing the evolving nature of VVS representations for supporting adversarial robustness.

A related limitation stems from the fact that, despite the scale of the NSD, neural responses cannot be straightforwardly pooled across individuals. Substantial anatomical and functional variability across brains makes pooling, or hyperalignment (Haxby et al. (2020)), challenging. As a result, only single subject's data could be leveraged for training neural predictors, which further limits the fidelity of the signal used for guidance. Recent advances in large scale brain foundation models (Wang et al. (2025)) that allows building shared representational spaces across individuals offer a promising direction for overcoming this constraint and allow future work to more fully probe the potential of human visual representations.

Interestingly, adversarial attacks on the human visual system have achieved some success, particularly when humans are limited to brief viewing times, reducing the likelihood of engaging feedback connections that most DCNN architectures lack (Veerabadran et al., 2023; Gaziv et al., 2023). Moreover, monkey IT neurons' preferences have been shown to be manipulable by adversarial attacks (Guo et al., 2022). Therefore, while our findings affirm that the human visual system likely operates in a representation space with geometries more conducive to robustness, the human visual system may not be entirely impervious to such attacks. In particular, in our experiments, even the best-performing model does not enable DCNNs to approach the expected robustness of human performance, as these almost imperceptible adversarial perturbations should not affect human perception at all. This discrepancy could certainly reflect limitations inherent to the quality of fMRI signals as discussed above, the neural guidance training pipeline, or the choice of ROIs, but it may also indicate that the human visual system itself is not the ultimate gold standard. We leave the investigation into the limits of human visual robustness to future work.

## 6 ETHICS STATEMENT

This work makes use of human fMRI data from the Natural Scenes Dataset (NSD) (Allen et al., 2022) openly accessible to the research community. We did not collect any new human subject data ourselves, and we have clarified the source and usage of this dataset within the manuscript.

## 7 REPRODUCIBILITY STATEMENT

We have described all methods, model architectures, training procedures, and experiments in detail in the main text and supplementary materials. In addition, we provide an anonymized code repository zip as part of the supplementary materials, which allows reproducing all experiments and results presented in the paper.

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

## A  Selection of Brain Regions of Interest (ROIs)

We focused on ROIs spanning the VVS from early visual areas to higher-order object-selective cortices along the ventrolateral surface of the brain. Specifically, we selected seven ROIs: V1, V2, V4, VO (Ventral-Occipital), PHC (posterior Parahippocampal Cortex), LO (Lateral-Occipital), and TO (Temporal-Occipital) (illustrated in Fig. 1B).

V1 and V2 are retinotopically organized areas known for processing low-level details (Marr, 2010) and represent the initial stages of the cortical processing hierarchy. V4 serves as an intermediate stage. While not yet exhibiting a pronounced preference for objects, V4 demonstrates early signs of invariance (Rust & DiCarlo, 2010). Further along the ventral cortex lie VO and PHC. VO is positioned immediately anterior to V4 with a demonstrated object preference (Brewer et al., 2005), while PHC, located further anterior and overlapping with the Parahippocampal Place Area (PPA) starts to show preference for scenes (Arcaro et al., 2009), indicative of its capability to extract global information. LO on the lateral surface, which includes LO1 and LO2, overlaps with the relatively posterior section of the functionally defined Lateral Occipital Complex (LOC) (Larsson & Heeger, 2006), commonly recognized for its holistic shape processing capabilities (Grill-Spector et al., 1999). Finally, TO, although less frequently investigated on its own in relation to object recognition, is situated anterior to LO, and potentially encompasses the more anterior portion of LOC (Larsson & Heeger, 2006).

Importantly, we do not assume a strictly linear progression of robustness across all higher-order regions. In particular, based on the anatomical positions and functional roles described above, we expect V4 to confer greater robustness than V2 and V1, with all these three early visual areas trailing the higher-level regions. We also expect PHC to confer more robustness than VO due to its scene-level integration, and TO to outperform LO as a more anterior and potentially more abstract object-selective area. However, the relative contributions between the ventral (VO–PHC) and lateral (LO–TO) pairs remain an open question. Therefore, we expect a fork-like hierarchical pattern, where the ventral (VO–PHC) and lateral (LO–TO) ROI pairs may show parallel robustness trends, with a clear progression within each branch (see schematic illustration in Fig. 5). This pattern can indeed be observed across our replication experiments (lower rows in Fig. 7, 8, and 9).

## B  Neural predictor quality check

We conducted evaluation and control experiments to ensure the quality of the neural predictors used in training NG models. Importantly, we show results from three other NSD subject (Subject-2, 5, and 7) in addition to Subject-1 focused on in the main text, as their data were used for replication experiments in Appendix C.1. These four subjects were selected among the eight subjects included in the NSD, because of their completion of the full natural image viewing seccion in NSD, thus providing a more adequate dataset for training the neural predictors.

We evaluated prediction quality by computing Pearson's $r$ between predicted and actual neural response patterns for each ROI (see Fig.6). We show the noise ceiling, as vertical bars, estimated using split-half correlations with Spearman-Brown correction (Yamins et al., 2014), leveraging repeated

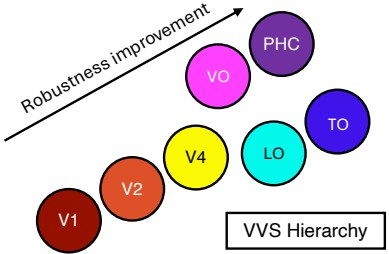

Figure 5: Schematic illustration of expected robustness improvements hierarchy across the seven ROIs included in our study.

image presentations in NSD. As can be observed, across all four subjects, neural predictors achieved performance approaching this noise-bounded upper limit.

To confirm that these correlations reflect meaningful learning rather than fitting noises in the fMRI data, we trained control predictors on shuffled data from Subject-1. Specifically, for each input image, we assigned a randomly selected, non-corresponding neural response as the ground truth. However, the integrity of the input-output correspondence was maintained during testing. Therefore, if the correlations can still be observed under these shuffling conditions, then this would suggest that the neural predictors are fitting neural-characteristic noises. However, as shown in Fig. 6, predictors trained on this shuffled data consistently performed near chance across all ROIs, validating that the original predictors learned informative, image-relevant representations.

We emphasize that these shuffled predictors were used solely for model validation and were discarded before our main analyses. The "V1-shuffle" and "TO-shuffle" conditions in the NG model experiments instead used predictors trained on valid neural data (see Methods in the main text for details) .

In addition, as shown in Fig. B, there are visible differences in both fMRI signal quality across regions (shown by split half correlations) and the performance of the neural predictors themselves (shown by Pearson's $r$). This raises a reasonable question about whether the hierarchical robustness pattern observed in the main manuscript simply reflects a hierarchy in fMRI signal quality or neural predictor fidelity. In other words, it is possible that TO guidance performed best because TO had the cleanest fMRI signal or the most accurate predictor.

To rule out this possibility, we tested the relationship between neural predictor performance and the robustness gains induced by neural guidance. Specifically, we computed the correlation between neural predictor performance as quantified by Pearson's $r$ as detailed above and the corresponding robustness $\Delta$AUC from Fig. 2A. The relationship was not statistically significant (Spearman's $\rho$ = 0.40, p = 0.29). We repeated the analysis using neural predictor performance normalized by the split-half correlation quantifying the fMRI signal quality in each region, to account for regional differences in fMRI signal SNR. This correlation was also not statistically significant (Spearman's $\rho$ = 0.23, p = 0.55). These results thus indicate that neither raw neural predictor accuracy nor fMRI signal quality could fully account for the hierarchical robustness improvement pattern.

## C  REPLICATING ROBUSTNESS IMPROVEMENT HIERARCHY FROM NEURAL GUIDANCE

### C.1  ACROSS ADDITIONAL NSD SUBJECTS

To assess the consistency of our findings across individuals, we trained NG-models using three additional NSD participants (Subjects 2, 5, and 7), each of whom, as described above, completed the full natural image viewing sessions and thus allowing sufficient data for training. We evaluated robustness improvements under untargeted $l_\infty$-based PGD attacks. As shown in Fig.7, all three subjects exhibited the same hierarchical trend observed in Subject 1 (main text Fig.2): models guided by later visual areas consistently demonstrated greater robustness gains than those guided by early regions.

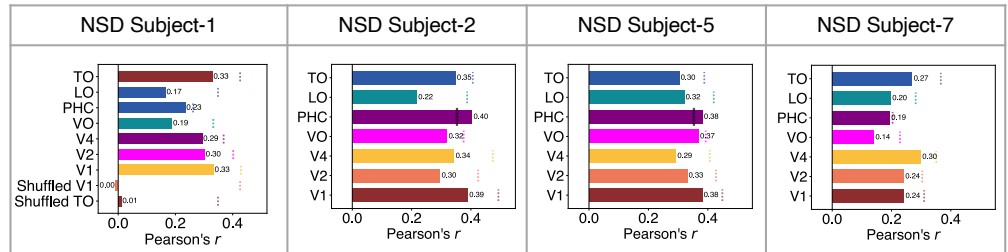

Figure 6: Quality checks of the neural predictors, shown as Pearson's r correlation between the predicted neural responses and the actual human recordings. Vertical bars indicate the noise-bounded upper limit of neural predictor performance, estimated using split-half correlations with Spearman-Brown correction (Yamins et al., 2014) for each ROI. In Subject-1, we included shuffled conditions ("Shuffled TO" and "Shuffled V1") served as baseline conditions to show that the neural predictos are indeed learning meaningful information (see Appendix B).

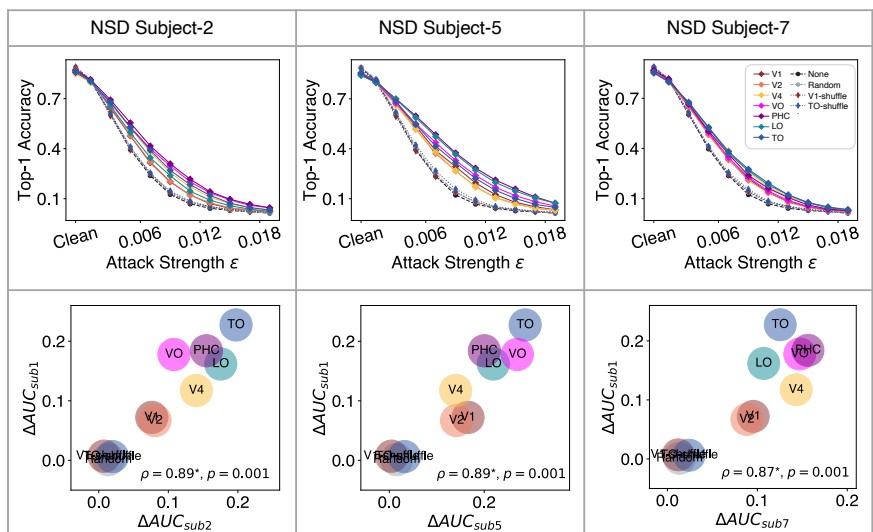

Figure 7: **Top**: $l_\infty$-PGD attack performance for NG-models guided by Subject-2, 5, and 7 in the NSD dataset. **Bottom**: The robustness improvement measured in $\Delta$AUC observed for these additional subjects (x-axis) show significant correlation with that of Subject-1 (y-axis), validating the results across individuals.

To quantify the consistency in the robustness hierarchy, we computed $\Delta$AUC between each NG model and the baseline (None) model and assessed the correlation in robustness hierarchies across subjects. Spearman's $\rho$ values showed statistically significant alignment with the hierarchy observed in Subject 1 (see the lower row of Fig.7). We note that the variability in magnitude is expected, potentially because of both individual differences and variances in the quality of neural data as well as neural predictors, as reflected by noise ceilings and predictor performance in Fig.6.

## C.2 ACROSS ADVERSARIAL ATTACKS

We further tested whether the hierarchical robustness improvement observed in NG-models generalizes across different types of adversarial attacks. Using NG-models trained on Subject 1, we evaluated their robustness under four additional attack benchmarks: 1. $l_\infty$-based AutoAttack (Croce & Hein, 2020), which aggregates several attack methods including FAB, Square Attack, APGD-CE, and APGD-DLR; 2. $l_\infty$-based Fast Gradient Sign Method (FGSM) (Goodfellow et al., 2014); 3. $l_2$-based Fast Gradient Method (FGM) attack (Goodfellow et al., 2014); and 4. $l_2$-based DeepFool attack (Moosavi-Dezfooli et al., 2016).

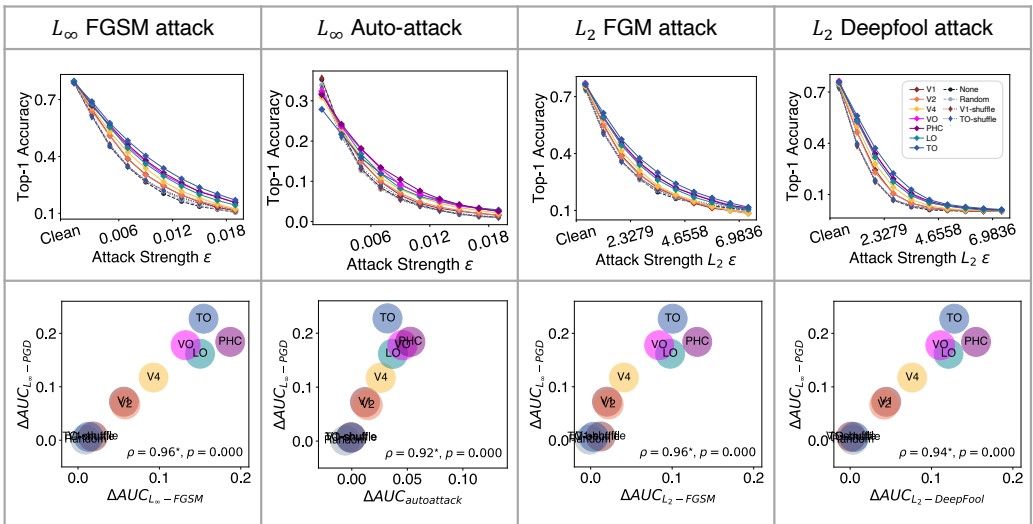

Figure 8: **Top**: Robustness of NG-models trained with neural data from Subject-1 under four alternative adversarial attacks. **Bottom**: Robustness improvement hierarchies under these attacks (x-axis) compared to the original $l_\infty$-based PGD results (y-axis) reported in the main text. Spearman's $\rho$ values indicate significant correspondence across attack types.

Despite variations in absolute accuracy, all attack types reproduced the same hierarchical trend: models guided by higher-level ROIs consistently exhibited greater robustness. We quantified this alignment in hierarchy by computing Spearman's $\rho$ between robustness gains ($\Delta$AUC) from each ROI under each attack and those under the $l_\infty$-based PGD attack reported in the main text. All correlations were statistically significant (Fig.8, lower row), confirming that the observed hierarchy is robust across a range of attack benchmarks.

### C.3 ACROSS ALTERNATE TASKS AND DATASETS

**CIFAR-100**    To assess the effect of neural guidance on a dataset with different visual statistics, we replicated our experiment using another classic classification benchmark, CIFAR-100 (Krizhevsky et al., 2009). We used Subject-1's neural data and trained seven additional NG-models, one for each ROI. Given the resolution mismatch with the NSD images ($227\times227$, $32\times32$ for CIFAR-100), we adapted both the neural predictors and NG-models: we modified the initial convolutional layer (Conv1) of both of the DCNNs by reducing both the kernel size (reduced from 7 to 3) and stride (reduced from 2 to 1) to better accommodate the smaller image size. For neural predictors, we trained them with MSCOCO images (those shown to the participants in NSD) that are downsampled to CIFAR-100 resolution for better generalizability.

The NG-models were then evaluated under the same $l_\infty$-based PGD attack. As shown in Fig. 9, while the overall robustness gains were reduced, potentially due to the limited generalization of neural predictors trained on MSCOCO images to CIFAR-100, the hierarchical pattern of robustness across ROIs was preserved. As in previous experiments, we observed a significant Spearman's $\rho$ between the $\Delta$AUC values and the VVS hierarchy.

**Image captioning with MSCOCO**    To test whether the robustness benefit of neural guidance generalizes beyond classification, we evaluated the NG-models on an image captioning task. Following standard practice, we used each NG-model's convolutional backbone (with both heads removed and weights frozen) as a feature extractor, and paired it with a standard LSTM-based decoder equipped with an attention module (Xu et al., 2015). The decoder was initialized with pre-trained weights and then fine-tuned on MSCOCO images viewed by Subject-1 in NSD (the same images used for training the neural predictors). This is to ensure that both the predictors and decoders operate on similar visual content to maximize the transferability. Evaluation was then performed using standard BLEU scores (Papineni et al., 2002). To assess robustness, we applied the same $l_\infty$-based PGD attacks used in the classification experiments.

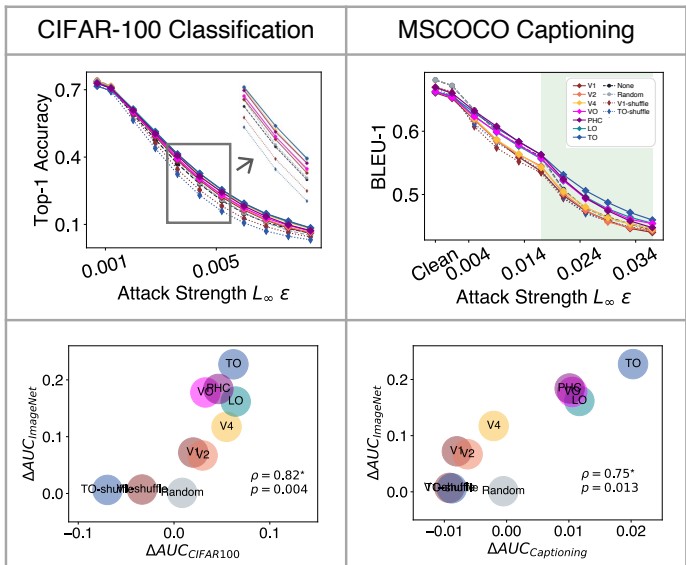

Figure 9: **Top**: Robustness of NG-models trained with Subject-1 evaluated on CIFAR-100 classification (left) and MSCOCO image captioning (right) tasks under $l_\infty$-based PGD attacks. Black squares and arrows indicate zoomed-in regions for visibility. For the captioning task, the shaded green region highlights the $\epsilon$ range where the hierarchical robustness pattern emerges. **Bottom**: Robustness improvement hierarchies (measured by $\Delta$AUC) from the two tasks (x-axis) compared to the ImageNet classification results (y-axis) reported in the main text. Significant correlations again confirm the replicability of the hierarchical effect across tasks and datasets.

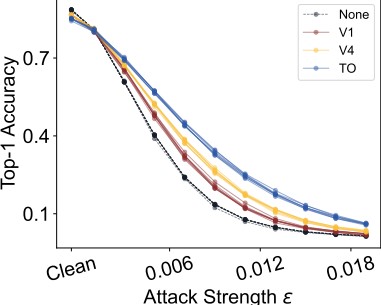

Figure 10: Reliability check across random seeds. We retrained four models (None, V1-guided, V4-guided and TO-guided) under six random seeds and evaluated them with the same $l_\infty$ PGD attack across the same epsilon bounds as in the main experiment. Curves from different seeds share the same color for each model and nearly overlap, thus indicating minimal variation.

As shown in Fig. 9, despite overall smaller effect sizes, likely due to the decoder being trained on frozen representations from a classification-focused model, we again observed a consistent robustness hierarchy. NG-models guided by higher-order ROIs produced captions that were more robust to adversarial perturbations, particularly at higher attack strengths (see green region in Fig. 9), mirroring the patterns observed in classification tasks. Therefore, we were able to show a replication of the robustness improvement hierarchy from neural guidance in the context of image captioning as well.

## C.4 ACROSS RANDOM INITIALIZATION OF MODELS

To confirm that the hierarchical robustness pattern observed is not a result of variability driven by random initialization of the model weights, we retrained a representative subset of models under six different random seeds. These models span the full range of the hierarchy, including the base-

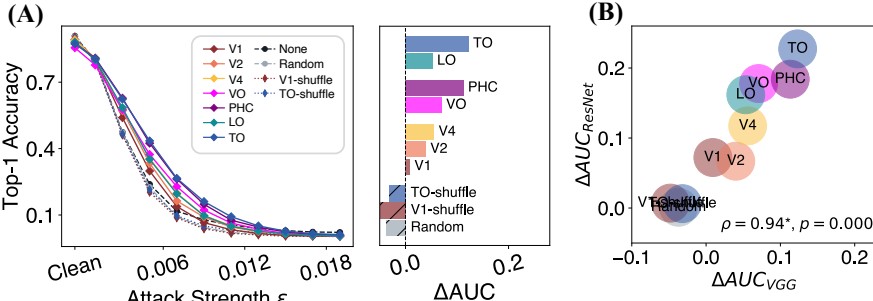

Figure 11: Replication of results on VGG-16. We trained 11 additional models (None, random, V1-shuffle, TO-shuffle, V1-guided, V2-guided, V4-guided, VO-guided, PHC-guided, LO-guided and TO-guided) using a VGG-16 based architecture rather than a ResNet architecture, and evaluated them with the same $l_\infty$-based PGD attack across the same epsilon bounds as in the main experiment. Both the accuracy curves and the resulting $\Delta$AUC values show a clear hierarchical improvement that mirrors our main results (**A**), and the $\Delta$AUC values correlate strongly with those obtained from the ResNet-18 based models (**B**).

line None model together with the V1-guided, V4-guided and TO-guided models. As shown in Figure 10, adversarial robustness varies only minimally across seeds for all models. The curves produced by different seeds for a given model remain nearly identical, and the ordering across regions is unchanged. These findings show that the robustness hierarchy reported in the main text is stable and not explained by random variation during training.

## C.5 ACROSS DCNN ARCHITECTURES

To verify that the hierarchical robustness improvement is not specific to the ResNet architecture used in the main experiments, we replicated our training and evaluation procedure on a different convolution-based model, VGG-16. Unlike ResNet-18, VGG-16 is much deeper and lacks skip connections, providing a complementary test of architectural generality. We trained 11 models covering the full set of baselines and neural guidance conditions used for ResNets.

As shown in Figure 11, we were able to closely reproduce the hierarchical robustness pattern. Models guided by higher level ROIs again showed stronger gains than those guided by earlier areas. One notable difference is that the random, V1-shuffle and TO-shuffle controls performed slightly worse than the plain baseline model. This could be a result of architectural differences as ResNets have skip connections that allow them to function as an ensemble of independent paths (Veit et al., 2016), which may enable them to better handle conflicting training objectives. Nevertheless, the improvements obtained from true neural guidance remain clear and follow the same hierarchically ordered progression observed in the main results.

## D    UNIQUENESS OF NG-MODELS IN OUTPUT SURFACES AND
    REPRESENTATIONAL SPACES

### D.1    WD-MODELS

To demonstrate that NG-models achieve a unique output surface and representational space that differs from conventional regularization techniques, we trained additional ResNet-18 models ("WD-models") on the ImageNet-50 dataset. These models were trained by varying the weight decay parameter (Loshchilov & Hutter, 2017), a common method for imposing regularization Rosca et al. (2020). For a better comparison, we selected five distinct weight decay values, targeting multiple robustness levels within a relatively comparable range comparable to those observed in neural-guided models, as shown in Fig. 12.

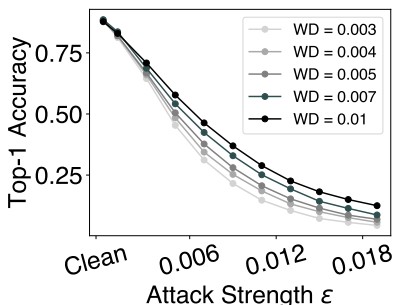

Figure 12: Robustness performance of WD-models trained with five different weight deays values to impose stronger regularization under $l_\infty$-based PGD attacks. The gradient of performance is comparable to those observed with Neural Guidance (Fig. 2A).

### D.2 SMOOTHNESS CHECK

A key characteristic observed of robust models is their association with smoother output surfaces or more slowly changing loss landscapes with respect to the input images Yu et al. (2019). Adversarial attacks exploit irregular and complex output surfaces, which facilitates the search for adversarial examples via gradient descent and creates numerous adversarial examples around each correctly classified data sample. Regularization methods such as the straightforward WD manipulation we here adopted (Loshchilov & Hutter, 2017; Rosca et al., 2020), along with other more sophisticated defenses Cohen et al. (2019) all leverage this insight to smooth the neural network function.

On the other hand, from a representational perspective smooth output surfaces may also arise as a natural consequence of operating on a well-structured representational space. In such a space, semantically similar images—including perturbed instances—are mapped to nearby locations, ideally lying on tighter and disentangled category manifolds (Chung et al., 2018; DiCarlo & Cox, 2007). Linear decision boundaries can then more easily separate category manifolds that include all identity-preserving transformations of object instances. Therefore, we should expect NG-models to also show smoother output surfaces if neural guidance induces more ideally structured representational geometry.

Given an input data $x$, ground truth $y$, and the loss function $\ell_\theta$, we first quantified models' smoothness loss (Yang et al., 2021) as the "worst" smoothness in a given data point $x$'s $l_\infty$-based neighborhood:

$$L_{\text{smooth}}(\ell, \theta, x, y) = \max_{\|\hat{x}-x\|_\infty} \|\nabla_{\hat{x}} \ell_\theta(\hat{x}, y)\|_2 \tag{6}$$

where $\hat{x}$ is the adversarial image obtained using PGD optimization. We then calculate the gradient norm of the loss with respect to $\hat{x}$. Note that for this quantification, higher values suggest larger gradients, steeper slopes, and therefore less smooth surfaces. For clarity, we applied the following transformation: $e^{-L_{\text{smooth}}}$ so that higher smoothness scores correspond to a smoother output surface.

As shown in Fig. 13B, WD-based regularization produced the expected monotonic increase in output surface smoothness with increasing WD magnitude. As predicted, NG models also exhibited enhanced smoothness compared to the baseline None model Fig. 13A. Interestingly, even the control models (random, V1-shuffle, TO-shuffle) showed moderate smoothing effects, consistent with the long observed benefits of noise injection (Gulcehre et al., 2016). However, all NG-models outperformed these baselines in smoothness across neighborhood radii, with later-region models exhibiting the greatest effects. This hierarchy also largely mirrored that of robustness improvement reported in the main text.

### D.3 TRANSFER ATTACK

To further assess the similarity of the loss landscape with respect to input images between NG models and the WD models trained with conventional regularization methods, we employed transfer attacks (Papernot et al., 2016) as an alternative evaluative measure. Specifically, adversarial examples were generated using the $l_\infty$-based PGD attack by attacking the standard "None" model, which

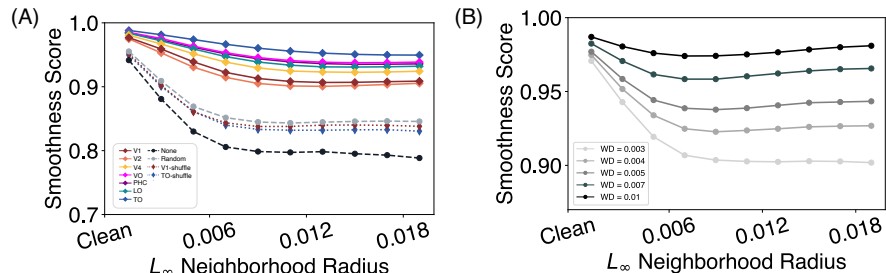

Figure 13: Output surface smoothness scores for (**A**) NG-models and (**B**)WD-models. Higher values denote smoother output surface.

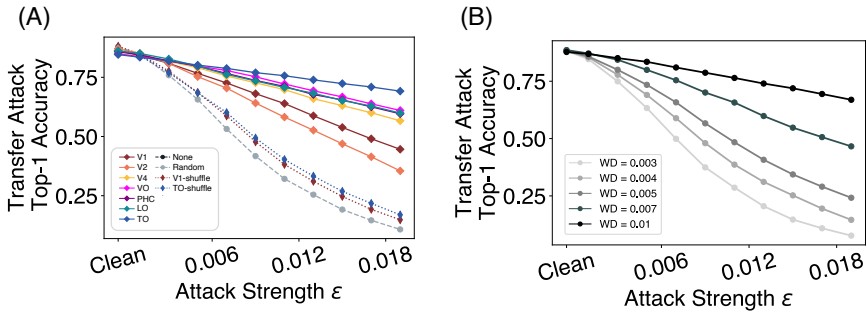

Figure 14: Transfer attack performance (assessed using adversarial examples generated from the None-model—a standard ResNet-18 DCNN trained on the ImageNet classification task only) for (**A**) NG-models and (**B**) WD-models.

had undergone only fine-tuning on the ImageNet-50 dataset. These adversarial examples were then applied to both the set of five WD-models and the seven NG models to evaluate their classification accuracy on these perturbed inputs. *Poor* accuracy then indicates compromised performance and thus a *greater* similarity in the loss landscape, or more precisely, the positioning of adversarial examples, between the source "None" model and the target model.

As expected, when attacking NG models using adversarial examples generated from the None model, neurally-guided models showed superior robustness compared to all baseline models (Fig. 14A), again with hierarchically later regions exhibiting increasingly better resistance to transfer attacks. Importantly, a similar increase in resistance to the transfer attack was also observed in WD-models (Fig. 14B) given that increased smoothness should generally lead to fewer vulnerabilities to be exploited. Such a result might tempt one to consider the hierarchical improvement in robustness observed in NG models as simply a reflection of the increasing regularization. Importantly, however, as we showed in the main text (Fig. 2B) with comparable and even lower native robustness, neurally-guided models consistently outperformed WD-models in resisting transfer attacks. Therefore, NG-models developed not only more smooth and thus robust output surface, but also uniquely different ones from conventional regularization effect.

### D.4 REPRESENTATIONAL SIMILARITY ANALYSIS (RSA)

We applied RSA (Kriegeskorte et al., 2008) to assess the similarity of representational spaces across all models and human ROIs. To enable direct comparison with human ROIs, we used the MSCOCO images presented during the NSD experiment to construct representational spaces. For all DCNNs, we extracted 1D feature vectors from the average-pooling layer for each image. We then computed pairwise Euclidean distances between all image features to form a Representational Dissimilarity Matrix (RDM) for each model. Subsequently, we computed Spearman's $\rho$ correlations between RDMs to produce a Representational Similarity Matrix (RSM), capturing pairwise similarities among the representational spaces of all models and human ROIs. To produce the visualization in Fig. 2C, we applied Multidimensional Scaling (MDS) to the RSM, embedding the models into a

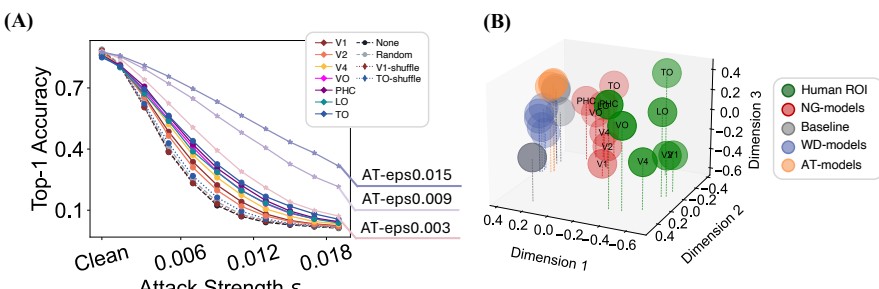

Figure 15: Comparison between neural guidance and adversarial training. **(A)** We trained three additional vanilla ResNet-18 models using single step adversarial training with $l_\infty$ perturbation bounds of $\epsilon$ equal to 0.003, 0.009 and 0.015. Their robustness performance under $l_\infty$-based PGD attacks across multiple bounds are shown in different shades of purple. **(B)** Representational similarity analysis, following the procedure used in Figure 2C, now includes the adversarially trained models shown in orange. Text labels for baseline models, weight decay (WD) models and adversarially trained (AT) models are omitted as these points cluster closely and in-plot labels are not readable.

three-dimensional space where models or ROIs with more similar representational spaces are positioned closer together.

### D.5 ADVERSARIAL TRAINING

To examine how our findings relate to the established defense approaches, in particular, adversarial training (Madry et al., 2017), we applied a single step adversarial training procedure using $l_\infty$-based PGD attack to generate perturbations for training. We trained three additional ResNet-18 models with perturbation strengths of 0.003, 0.009 and 0.015. As a result, these models showed substantial gains in adversarial robustness (Fig. 15A) when evaluated under the same $l_\infty$-based PGD attack with the same range of $\epsilon$s used in the main experiment. This is consistent with the effectiveness of adversarial training widely reported in prior work (Bai et al., 2021).

However, despite the superior performance, these adversarially-trained models do not approach the adversarial robustness problem in the same way as humans. To probe such mechanistic differences, we repeated our representational similarity analysis and included these adversarially trained models. The resulting geometry (Fig. D.5B) showed that adversarially trained models cluster tightly with the baseline and weight decay models. In contrast, as shown and discussed in the main text, the neurally guided models moved toward the representational geometry of human VVS regions. Such difference is consistent with the observation on the weight decay controls, and highlights that human inspired representational geometry offers a uniquely different direction for robustness that differs from these engineering driven defenses.

## E NEURAL CATEGORY MANIFOLDS

### E.1 SELECTION OF IMAGES FOR ANALYZING HUMAN NEURAL MANIFOLDS

To estimate category manifold properties, extent and linear separability, using the MFTMA framework (Chung et al., 2018), we need multiple representative samples per object category. For NG-models trained on ImageNet-50, this process was straightforward: we used the test set with 50 categories and 100 samples per category (50 original and 50 perturbed images). However, for human neural data from the NSD, selecting appropriate category-level samples was more challenging. NSD participants viewed naturalistic images from the MSCOCO dataset (Lin et al., 2014), which is curated for tasks such as object detection and segmentation and often contains complex scenes with multiple overlapping objects (see Fig. 16A for an example), making them unsuitable for object-centric manifold analysis.

To address this, we curated a subset of images viewed by Subject-1 that were more appropriate for representing single-object categories. Our filtering procedure was as follows: For each image, we

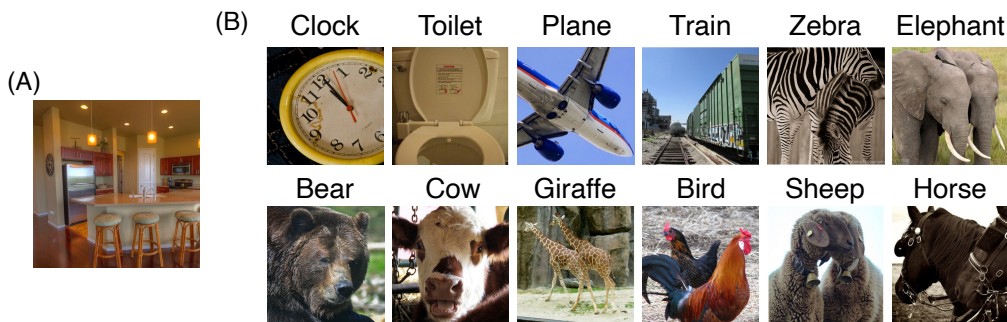

Figure 16: (**A**): Example MSCOCO image presented to NSD participants containing multiple prominent objects, making it unsuitable for category-level manifold analysis. (**B**): The 12 object categories selected for human neural manifold analysis using MFTMA, each illustrated with an example image containing a single, dominant object.

identified whether it contained only one object category or whether one object category occupied a significantly larger area (based on aggregated bounding box area) than all other objects combined. If either of the conditions was satisfied, we assigned the image to the dominant object category. Within each selected category, we ranked the images by object size (bounding box area aggregated across all instances of the dominant object category) in descending order and retained the top 50 images per category to serve as category manifold samples. This process yielded a final set of 12 object categories with more appropriate single-object exemplars for MFTMA (see Fig. 16B for the selected categories and representative examples).

### E.2 CATEGORY MANIFOLDS VISUALIZATION

To visually illustrate how category manifolds are transformed across the human VVS, we applied Isomap to neural representations of three example categories (elephant, clock, and train) selected from the 12 filtered categories described above. We performed this separately for V1, V4, and TO, representing early, intermediate, and higher-order regions in the VVS, respectively (see Fig. 3A).

Following the manifold disentangling hypothesis (DiCarlo & Cox, 2007), which posits that identity-preserving transformations of objects form continuous and separable manifolds, we treated each category as an independent manifold. Isomap was thus applied independently to each category's high-dimensional neural representations to obtain their 3D embeddings. However, since Isomap embeddings give arbitrary rotation and translation, we registered all manifolds into a common visualization space so that we can compare their relative position, extent and orientation. Specifically, we first extracted a shared 3D basis using the original high-dimensional neural representations with Principal Component Analysis (PCA). We then projected all representations into this common space, and they serve as targeted references for alignment. For each category, we used orthogonal Procrustes analysis to find the optimal rotation that aligned its Isomap embedding to the corresponding projection in the common space. This ensured that manifold shape and internal geometry obtained using Isomap were preserved while aligning all manifolds into the same coordinate system. Finally, each aligned manifold was translated to match the centroid of its corresponding PCA-projected reference and rescaled to match its average spread, thus enabling meaningful visual comparison of scale and spacing.

This visualization illustrates that category manifolds become increasingly compact (smaller extent, as reflected by the decreasing axis ranges in Fig. 3A) and appear more separated from one another in the higher-order visual area TO compared to others. We note that this analysis is intended for illustrative purposes only. For the quantitative assessment of overall manifold properties, we refer readers to the MFTMA-based analyses reported in the main text.

## F    ADVERSARIAL VERSUS OUT-OF-DISTRIBUTION (OOD) ROBUSTNESS

Here, we clarify why our work focuses on adversarial robustness rather than the broader space of OOD corruptions which are typically non-gradient-based and unbounded. Specifically, the heterogeneous findings regarding OOD robustness benefits from bio-inspired models in prior work and the specific theoretical needs for our manifold analysis make adversarial attacks a more natural choice.

Prior work on neural alignment has shown that while alignment confers adversarial robustness, its benefits for broad OOD benchmarks are often heterogeneous. For instance, Safarani and colleagues (Safarani et al., 2021) found that V1-aligned models improved robustness only against specific noise types, while sometimes performing worse on others. Interestingly, even bio-inspired models with architectural changes beyond representational alignment like VOneNet also show performance degradation on certain OOD transformations (Malik et al., 2023).

We argue that this heterogeneity highlights two critical implications that motivate our focus on adversarial robustness:

First, Such mixed findings suggest that "OOD robustness" is less likely a unitary phenomenon and may not have a straightforward link to representational geometry. Instead, it could be a broad umbrella comprising diverse transformations that likely recruit distinct, specialized cognitive mechanisms. For example, handling occlusion may require contour completion that necessitates feedback signals from higher-order brain regions (Muckli et al., 2015), and recognizing extreme viewpoints may require effortful mental rotation (Shepard & Metzler, 1971). In contrast, adversarial robustness has been explicitly shown a link to manifold geometry, as prior work (Dapello et al., 2021) has demonstrated that adversarially trained models seem to develop more compressed and linearly separable manifolds.

Second, our study tests the manifold disentangling theory where one essential claim is that category manifolds should consist of identity-preserving transformations. However, humans are not fully foolproof for these OOD transformations, showing performance degradation just like the DCNNs on transformations like adding uniform noises, phase noises or even changing contrast (Geirhos et al., 2018). Since human performance on many OOD benchmarks remains uncharacterized, they are less ideal for strictly characterizing manifold geometry. On the other hand, the nature of almost imperceptibility for adversarial perturbations safely ensures that they mostly remain "on-manifold" for humans, making them the more precise choice given our focus on these manifold structures.

That being said, determining how far simply aligning representational geometry can go with OOD transformations and what additional mechanisms are required is indeed an interesting and important direction for future work, but falls outside the scope of the present study.

## G    VISUALIZATION OF ADVERSARIAL PERTURBATIONS

To provide a concrete visual reference for the attack strengths used in our experiments, we visualize sample adversarial images in Figure 17 across three different epsilons used in our main experiment. As shown, perturbations at the lower end ($\epsilon = 0.003$) are virtually invisible under standard viewing conditions. At a higher strength of our experimental range ($\epsilon = 0.015$), noise artifacts become readily detectable upon careful inspection, presenting as high-frequency colorful texture patterns.

Critically, prior work (Elsayed et al., 2018) has demonstrated that to make these perturbed images readily perceivable and successfully deceive human observers with adversarial attacks, perturbation magnitudes need to exceed $\epsilon > 0.1$ (approximately $32/255$). Our maximum perturbation strength ($\epsilon = 0.019$) remains nearly an order of magnitude below this human robustness threshold. Thus, while the noise at higher strengths may be perceptible, the images safely remain "on-manifold" (identity-preserving) for human perception, highlighting the stark contrast with DCNNs, whose performance degrades significantly even at these sub-human-threshold magnitudes.

## H    LLM USAGE DISCLOSURE

Large language models (LLMs) were used only to assist with grammar, spelling checking, and minor language polishing.

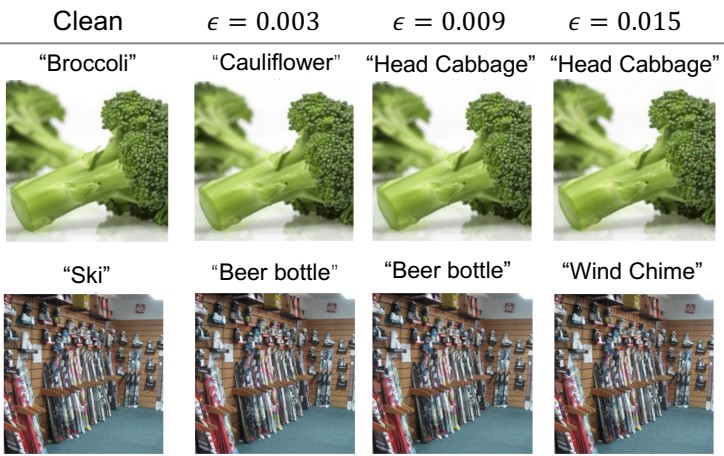

Figure 17: Visualization of adversarial examples generated using $l_\infty$-based PGD attacks at varying perturbation strengths ($\epsilon$). At the lowest strength ($\epsilon = 0.003$), the perturbation is effectively imperceptible. At the highest strength ($\epsilon = 0.015$), high-frequency noise patterns become visible upon close inspection (e.g., texture-like colorful artifacts on the broccoli stem and background). Original and model-predicted (incorrect) labels are provided for each instance.

