# OpenReview forum: "Probing Human Visual Robustness with Neurally-Guided Deep Neural Networks"
_ICLR.cc/2026/Conference — Submitted to ICLR 2026_

### Official Review · Reviewer_8mca · 2025-10-29

**Soundness:** 3
**Presentation:** 4
**Contribution:** 4
**Rating:** 8
**Confidence:** 3

**Summary:**

This paper explores the connection between robustness and human neural (fMRI) responses, demonstrating that guiding model representations towards neural responses improves robustness. Furthermore, the authors expand on this result, creating a supervision metric that encourages the learned manifold to have human-inspired beneficial geometric properties. The authors demonstrate that this manifold guidance also improves robustness without the need for human data.

**Strengths:**

This paper is an important computational exploration of an outstanding hypothesis in visual neuroscience (that robustness is an emergent property of human vision and the VVS).
The authors have demonstrated exceptional rigor with comprehensive analyses for each experiment with all appropriate controls for each of the experiments presented.
The paper goes beyond demonstrating correlation of neural guidance with robustness to introduce "Manifold Guidance" as a form of supervision that isolates this geometric constraint, confirming that representational geometry in a model is sufficient to transfer the robustness principle. A new tool in the toolbox for inducing robustness will be useful, and it will be interesting to see what other aspects of human alignment emerge from this type of supervision.
While the main results are for a single subject, the authors analyze 2 subjects in the supplemental demonstrating generality.
The authors test the results over multiple attack types demonstrating generality.
The authors clearly state (all but one of) the limitations of their experiments
I find the paper overall exceptionally well written and generally easy to follow. The discussion in particular is interesting and contextualizes the results well.

**Weaknesses:**

The main weakness of the paper is in overstatement of implied architecture generality of the robustness findings reported. The evidence is based solely on results from ResNet-18. While ResNet is an obvious choice, adversarial robustness is highly dependent on architecture, so the claim that this is a result applying to DNNs broadly is unsubstantiated. If the authors wish to make this claim broadly, they should demonstrate if this result holds for other DNN architectures such as VIT, bio-inspired CORnet (which the authors mention in related work), and a simpler architecture such as VGG-16. Alternatively, the authors could restrict their claims (including in the title) to be regarding only ResNet-18
While the ordering of the robustness results (V1 to TO) is highly compelling and supports the central hypothesis, the magnitude of the quantitative improvement for manifold guidance (Figure 4) appears modest. The core finding currently rests on the trend of the means, not the statistical significance of the differences. This could be addressed by measuring variability across initialization seeds, and/or a statistical test for linear trend across regions.

Minor Points:
Formatting figures 5, 10

**Questions:**

I suggest the authors either:
1) Reduce the scope of the claim including in the title to claim these results only for ResNet, not DNNs generally
or
2) Test the results presented on a set of other model architectures including VIT, bio-inspired, and a simpler CNN model (no skip-connections).

---

> ### Author Response · Authors · 2025-11-21
> **Responses to reviewer 8mca**
>
> We thank the reviewer for these thoughtful comments. We address each of the reviewer’s concerns below.
>
> 1. **Use of only ResNet architectures**
>
>     We thank the reviewer for raising this critical point on generalizability. As also noted in General Responses 2, we have added additional experiments on VGG-16.
>
>     We first would like to clarify that we chose a lightweight ResNet-18 mainly for practical reasons. Our study requires training and experimenting with multiple models in order to characterize different stages across the ventral stream. Since our aim is not to develop a defense method where cross-architecture applicability is essential, ResNet-18 offers an efficient architecture that enables extensive analyses and serves well for a proof-of-concept investigation.
>
>     We agree, however, that our hypothesis suggests the effect should not be specific to ResNets. Therefore, we have also trained an additional set of VGG–16 models and observed the same hierarchical robustness pattern (see updated results in Appendix C.5 and Fig.11), indicating that the effect indeed generalizes beyond ResNet. However, due to the limited time for the discussion period, we may not be able to finish testing other architectures such as ViT. We leave this up to future work.
>
>     Bio-inspired models such as the CORnet architectures, however, are difficult for evaluation because they have explicit assumptions about correspondences between particular layers and brain regions. If we respect these assumptions and align, for example, V1 to the CORnet-V1 layer and V4 to the CORnet-V4 layer, this would change both the neural target and the model depth simultaneously, making it difficult to isolate the effect of guidance from architectural priors. On the other hand, ignoring these built-in correspondences would undermine a key aspect of what makes such architectures biologically informed. For this reason, we believe that generic convolutional networks provide a cleaner experimental platform for our goal of manipulating representational geometry at the end layer without imposing architectural assumptions.
>
>     As the reviewer suggests, we have constrained the scope of claim in line with the models tested. Rather than DNN, we revised the manuscript to use “deep convolutional neural networks (DCNNs)” .
>
> 2. **Reporting of statistical variation**
>
>     Thank you for highlighting this point. We agree that it is important to demonstrate that our main findings are stable across random initialization of the models. To assess this, we ran multiple random seeds for a representative subset of models, including the baseline None model, and the V1-, V4-, and TO-guided models to assess the reliability of our core hierarchical improvements shown in the main manuscript. These models span the entire hierarchy and therefore provide a meaningful test of whether random initialization affects the observed robustness pattern. As shown in the newly added Appendix C.4 and Fig.10, their adversarial robustness varies minimally across six additional random seeds used, suggesting that the hierarchical trend is not driven by random variance. We hope that the consistency observed across this representative set provides some reassurance about the reliability of the overall results. We have included these analyses in the revised manuscript.
>
> 3. **Minor formatting issues**
>
>     We appreciate the reviewer for pointing this out. We have cleaned up the formatting in the revised manuscript.

---

> > ### Comment · Reviewer_8mca · 2025-11-25
> > **Response to Authors**
> >
> > I thank the authors for their prompt response, and for including some of the additional experiments I recommended.
> >
> > The additional experiments included on VGG-16 address the need for testing these claims across different model architectures.
> >
> > The authors have also refined their claims throughout the paper to apply to CNNs, not deep networks generally, and now match the experiments that have been shown.
> >
> > I do not need any further clarification. I believe this is a strong paper, which has been further improved upon in response to other reviewer's comments as well, and I would argue for acceptance.

---

### Official Review · Reviewer_gx3G · 2025-10-31

**Soundness:** 2
**Presentation:** 3
**Contribution:** 2
**Rating:** 2
**Confidence:** 4

**Summary:**

The authors co-train DNNs for image classification and alignment with fMRI recordings from the human ventral visual stream. They found that alignment to higher-order regions of the visual system (i.e., LO vs. V1) yielded more linearly separable neural category manifolds and, potentially as a consequence, larger adversarial robustness gains in the DNNs. These results suggest that the representational geometry shaped through transformations applied by the ventral visual system are a key reason why humans do not experience the same sensitivity to adversarial perturbations as DNNs.

**Strengths:**

- Lots of interesting ideas and I really like the manifold analyses. It's an elegant way of comparing human and machine vision.
- Well done adding the weight decay controls.
- Very creative work.

**Weaknesses:**

- I'm struggling with the logic. Why would training on higher order layers lead to lower clean accuracy? I would actually guess the opposite. Higher order regions should encode more invariant object representations, which should train models to similarly have more tolerance to transformations as well as noise.
- No discussion about the inherent limitations of doing this with fMRI. How could that be a limiting factor for this work?
- "Past theories suggest human visual robustness arises from a
representational space that evolves along the ventral visual stream (VVS) of the
brain to increasingly tolerate object transformations" This is a weak sentence. And the follow up is at best underspecified, even for the abstract "To test whether robustness is
supported by such progression as opposed to being confined to specialized higher-
order regions..."

Minor: Line 220: "We first applied this analysis to human neural representations in each of the seven ROIs using NSD." I would use something like voxel activity patterns instead of human neural representations here. The latter is some latent property of the patterns that you hope to extract from the former.

**Questions:**

Questions:
- It would be good to know if any of the attack strengths are perceivable. Can you add that info?
- In the MDS: Any idea of what the dimensions capture?
- Why use regression slopes in Fig 3 but spearman in Fig 4? We should use spearman in both.
- "To further test the manifold hypothesis (DiCarlo & Cox, 2007) that identity-preserving transformations should be treated as part of the same category manifold, we enriched each category with adversarially perturbed variants of the test images, thus obtaining 100 samples in total per category, similar to previous work." The focus of the DiCarlo and Cox work was on identity-preserving transformations like translation and scaling. What about using these types of transformations? The more I think about it, the more confused I am that the authors focused on adversarial perturbations instead of naturalistic transformations for which there are a variety of popular benchmarks in computer vision (objectnet, imagenet-A, etc.).
- What about an adversarially trained model as a control?

---

> ### Author Response · Authors · 2025-11-21
> **Responses to reviewer gx3G**
>
> We thank the reviewer for their valuable comments. Below are our point-to-point responses.
>
> ## Weaknesses
>
> ### 1. I'm struggling...
>
> We thank the reviewer for raising this important point. We appreciate this opportunity to clarify our results.
>
> First, we wish to state clearly that we do not claim the TO-guided model is superior to V1-guided or vanilla models in all aspects. Rather, this observed trade-off between clean accuracy and robustness is critical evidence showing that neural guidance, and neural guidance from higher-order regions, effectively steers the model towards more human-like features, specifically, those that are robust but less optimized for exploiting the brittle statistical shortcuts of ImageNet.
>
> This interpretation is grounded in a substantial body of literature [1-3]. This line of past work has shown that, under supervised classification, there is an inherent tension between optimizing for standard accuracy and robustness. In naturalistic datasets such as ImageNet, many cues are highly predictive, but at the same time not stable under perturbations (e.g. cars must have roads in the background to be cars). Models are trained to maximize accuracy and therefore tend to rely on non-robust features that happen to correlate strongly with the label. Robust training methods, including the popular adversarial training (AT) that multiple reviewers mentioned, reduce dependence on these brittle cues by introducing uncertainty around them. However, this necessarily sacrifices the “easy” accuracy gains these cues provide.
>
> Our results here are thus consistent with these past interpretations. Alignment to higher-order VVS regions appears to shift the model towards more robust features (e.g., holistic shape) and away from non-robust texture or background cues. Consequently, the model loses the accuracy boost from those brittle features while gaining robustness.
>
> Crucially, however, while this trade-off mirrors engineering approaches like AT, Neural Guidance achieves it through a fundamentally different mechanism. As shown in our new Appendix D.5 (Fig. 15) and will be discussed in detail in our responses to Question #5 below, AT models do not align with human representational geometry. This suggests that while both methods shift feature reliance, Neural Guidance does so fundamentally differently by, for example, imposing representational geometric constraints. We also note that this pattern aligns with prior biological alignment work [4], which observed a similar trade-off when aligning models to mice’s V1 representations.
>
> We have also added the interpretation of this observation in the main results in section 4.1 of the revised manuscript.
>
> [1] Yang et al., 2020 A closer look at accuracy vs. robustness. NeurIPS, 33
>
> [2] Tsipras et al., 2018 Robustness may be at odds with accuracy. arXiv:1805.12152.
>
> [3] Ilyas et al., , 2019 Adversarial Examples Are Not Bugs, They Are Features. NeurIPS, 32
>
> [4] Li et al., 2019 Learning from brains how to regularize machines. NeurIPS, 32
>
> ### 2. No discussion...
>
> Thank you for raising this important point. We agree that fMRI comes with inherent limitations in signal quality and BOLD is a more indirect measure of neural activity than methods like intracranial recordings [1]. The necessary intermediate step of training neural predictors likely further introduces additional signal loss. These factors constrain how much information remains after being transferred from the human brain to the model and could contribute to the modest absolute robustness gains in our results.
>
> However, we would like to note that fMRI offers unique strengths that are essential for the present study. That is, it allows simultaneous access to multiple ventral stream regions within the same human participants at a fine-grained spatial scale. This property makes it possible to examine representational progression across the ventral stream within a unified experimental framework. While invasive methods like human ECoG offer higher resolution and direct measurement, they are typically constrained by limited spatial coverage and dictated by clinical needs, making such systematic, end-to-end comparisons less feasible.
>
> Furthermore, we relied on the NSD [2], a large-scale 7T fMRI dataset which provides data from participants viewing a large (close to 10k) and diverse set of natural images. This is a data scale critical for training DCNNs that is rarely achievable for other neuroimaging data obtained from humans.The high field strength, the extensive data collected and the advanced preprocessing pipeline used in this dataset provide one of the best currently available human vision neural dataset.
>
> We have added a discussion for the limitation of fMRI data in the Discussion section.
>
> [1] Logothetis 2008 What we can do and what we cannot do with fMRI. Nature
>
> [2] Allen et al., 2022 A massive 7T fMRI dataset to bridge cognitive neuroscience and artificial intelligence. Nat Neurosci

---

> > ### Author Response · Authors · 2025-11-21
> > **Responses to reviewer gx3G**
> >
> > ### 3. "Past theories...
> >
> > Thank you for raising this important point about clarity. Our intention was to convey that past influential neuroscience work has proposed that stages of the VVS progressively build increasingly complex and invariant object representations [1][2]. However, direct characterization of how such representations evolve across multiple VVS stages in humans has been limited, with prior work comparing only a few regional comparisons (e.g., V4 vs. IT [3]). As a result, it remains possible that robustness could be solved in or functionally localized to specialized higher-order regions in IT, rather than emerging gradually along the full hierarchy.
> >
> > We agree that our original phrasing did not make this distinction sufficiently clear. We have revised the abstract accordingly to make the intended contrast explicit: “... Past work in Neuroscience proposes that the ventral visual stream (VVS) builds an increasingly complex and robust object representation across successive stages. However, this hierarchical refinement across VVS has not been systematically demonstrated in humans, and it remains possible that robustness arises only in specialized higher-order regions such as IT. To distinguish these possibilities …”.
> >
> > [1] Riesenhuber & Poggio 1999 Hierarchical models of object recognition in cortex. Nat Neurosci
> >
> > [2] Serre et al., 2007 A feedforward architecture accounts for rapid categorization. PNAS
> >
> > [3] Rust & DiCarlo 2010 Selectivity and tolerance (“invariance”) both increase as visual information propagates from cortical area V4 to IT. Journal of Neurosci
> >
> > ### 4. Minor: Line 220...
> >
> > Thank you for pointing this out. We agree that “voxel activity patterns” is a more precise description than “human neural representations” at this stage of the analysis. We have updated the manuscript accordingly.
> >
> > ## Questions
> >
> > ### 1. It would be good...
> >
> > For a more concrete reference for perceptibility, we have added Appendix G and Figure 17, which visualizes sample adversarial images across the three perturbation strengths used in our experiments. As shown in the figure, perturbations at the lower end (epsilon=0.003) are effectively imperceptible under standard viewing conditions. At a higher strength used (epsilon=0.015), high-frequency noise patterns become visible upon close inspection (texture artifacts on the broccoli and ski images).
> >
> > We note that in order for the adversarial perturbation to be readily detectable and can fool human observers, prior work [1] has used perturbation magnitudes above 0.1 (~32/255). Our maximum strength ($\epsilon$=0.019) is nearly an order of magnitude lower than this threshold. Thus, while the noise at higher strengths is perceptible to the human eye, the images safely remain “on-manifold” (identity-preserving) for human perception, highlighting the stark contrast with DCNNs, whose performance breaks even at these lower magnitudes.
> >
> > [1] Elsayed et al., 2018 Adversarial examples that fool both computer vision and time-limited humans. NeurIPS 31
> >
> > ### 2. In the MDS...
> >
> > We would like to clarify that in our use of MDS, the primary goal was to visualize the overall similarity relationship among the different models’ and humans’ representational spaces, rather than to interpret the individual dimensions. MDS is a widely used method that provides a low dimensional embedding that preserves pairwise distances, allowing us to illustrate how neurally guided models cluster more closely with the human spaces relative to the vanilla models. In other words, we did not intend for the MDS axes themselves to correspond to psychologically or neuro-scientifically meaningful factors, but merely using it as a straightforward choice of visualization tool.
> >
> > ### 3. Why use regression...
> >
> > Thank you for pointing out this important difference. In Fig. 3, we used linear regression because we specifically intended to examine whether manifold statistics quantitatively predict the magnitude of robustness improvement. In this context, a regression slope provides a straightforward way to assess the strength and direction of a predictive relationship between the manifold extent or separability and the change in robustness, both of which are continuous values.
> >
> > In contrast, Fig. 4 focuses on whether the hierarchical ordering of robustness improvement is preserved between neural and manifold guidance, which is inherently a rank correlation based question. We therefore used Spearman’s rho correlations here and all the replication results such as the ones in App. C.
> >
> > However, we understand the reviewer’s concern. To confirm that our conclusions are not sensitive to the choice of statistical measure, we additionally computed Spearman correlations for the relationships shown in Fig. 3. The Spearman results are significant (extent vs robustness AUC: $\rho$=-0.79*, p=0.006; Capacity vs. robustness AUC: $\rho$=0.77*, p=0.009) and support the same conclusions as the regression analysis.

---

> > > ### Author Response · Authors · 2025-11-21
> > > **Responses to reviewer gx3G**
> > >
> > > ### 4. "To further test...
> > >
> > > We thank the reviewer for raising this important point. We can appreciate why this may have been confusing. We have briefly discussed in General Responses #3, and here we expand on the details.
> > >
> > > **Regarding why we use adversarial examples to probe category manifolds**: We would first like to clarify that our focus on the manifold hypothesis [1] is on the progressive disentangling of manifolds across the VVS. The factor that all identity-preserving transformations of an object should lie on the same manifold is certainly one of the most important claims in the original formulation of the hypothesis. However, we argue that the question of, concretely, which naturalistic transformations (or more generally OOD (Out-Of-Distribution) transformations) truly remain on a category manifold still requires more comprehensive empirical evaluations that are outside of the current scope.
> > >
> > > One reason for us to be cautious is that human performance on many naturalistic transformations, particularly those included in popular OOD benchmarks, has not been fully characterized and may not be uniformly strong. For example, prior work [2, Figure 3] has shown that humans can perform as poorly as DCNNs on certain distortions, such as added uniform or phase noises and even contrast changes. For more naturalistic benchmarks such as ObjectNet and ImageNet-A, many stimuli visually appear to be analogous to atypical or non-canonical views of objects, for which humans are known to require more effort and time to recognize the object [3]. Therefore, it is unclear which of these transformations, and under what magnitudes, can be safely assumed to lie on the same manifold as clean images. In contrast, adversarial perturbations present as more natural choices for constructing the manifold. These almost imperceptible perturbations more than likely do not affect human behaviors at all and therefore are more likely to be “identity-preserving”, thus remaining on or at least lying near the underlying category manifold. Therefore, we view adversarial perturbations as a natural choice for examining manifold structure in the context of our study.
> > >
> > > **Regarding why we broadly focus on adversarial robustness**: We would like to note that our central question concerns whether and how the representational geometry across the human VVS supports robustness. Adversarial robustness has been linked to the representational geometry, in particular, manifold disentangling, of models in prior work that shows better separability and more compressed manifolds as a result of adversarial training [1]. In contrast, OOD corruptions such as those in ImageNet-C span a quite heterogeneous set of challenges, and many of which likely engage complex downstream mechanisms. For example, handling occlusion may require contour completion involving feedback from higher-order regions [4], and recognizing objects under extreme viewpoint changes may require effortful mental rotation [5]. Therefore, the relationship between OOD robustness and VVS representational geometry is less clear. Determining how far simply aligning representational geometry can go with OOD transformations and what additional mechanisms are required is an interesting and important direction for future work, but falls outside the scope of the present study.
> > >
> > > We have also discussed these reasonings in detail in the newly added Appendix F to clarify our focus. We have the citations with more appropriate ones: (1) the computational disentangling framework [6] for the methods, and (2) one follow-up study [7] showing that adversarially robustified models show improved manifold statistics thus supporting a link between adversarial robustness and the manifold geometry.
> > >
> > >
> > > [1] DiCarlo & Cox 2007 Untangling invariant object recognition. Trends in Cog. Sci.
> > >
> > > [2] Geirhos et al., 2018 Generalisation in humans and deep neural networks. NeurIPS 31.
> > >
> > > [3] Beck et al., 2024 The Role of Real-World Statistical Regularities in Visual Perception. Current Directions in Psych. Sci.
> > >
> > > [4] Muckli et al., 2015 Contextual feedback to superficial layers of V1. Current Biology
> > >
> > > [5] Shepard et al., 1971 Mental rotation of three-dimensional objects. Science
> > >
> > > [6] Chung et al., 2018 Classification and Geometry of General Perceptual Manifolds. PRX
> > >
> > > [7] Dapello et al., 2021 Neural Population Geometry Reveals the Role of Stochasticity in Robust Perception. NeurIPS 34

---

> > > > ### Author Response · Authors · 2025-11-21
> > > > **Responses to reviewer gx3G**
> > > >
> > > > ### 5. What about an adversarially trained model as a control?
> > > >
> > > > We thank the reviewer for raising this important point. As also discussed in General Response #2, we have added additional experiments to show that adversarial training (AT) presents as another engineering-driven approach that is effective, but works through mechanisms different from humans. We expand on this below.
> > > >
> > > > We would first like to note that our goal is not to develop a new defense method that competes with other defenses, rather, use DCNNs as an experimental platform for testing hypotheses about what supports human’s robustness to adversarial perturbations. In this context, AT can be viewed as another engineering-driven intervention, similar to our weight decay controls. Both of them improve robustness, but neither induces representational changes that resemble the human ventral stream.
> > > >
> > > > To illustrate this distinction, we have trained three adversarially trained models with increasing perturbation bounds ($\epsilon$ = 0.003, 0.009, 0.015, see Appendix D.5, Fig. 15). As expected, AT produced substantial gains in robustness. However, importantly, these models showed representational geometries that remained very close to standard DCNNs and far from human VVS representations (Fig. 15B), paralleling what we observed for weight decay models. Neural guidance, expectedly, consistently produced a shift towards human ventral stream geometry.
> > > >
> > > > We have added these results in Appendix D.5, and a short discussion in section 4.2 of the main text.

---

> > > > > ### Comment · Reviewer_gx3G · 2025-11-25
> > > > > **Response**
> > > > >
> > > > > "use DCNNs as an experimental platform for testing hypotheses about what supports human’s robustness to adversarial perturbations"
> > > > >
> > > > > I like this idea a lot, and the use of representational geometries to probe it. How would one ultimately translate those differences between AT models vs. human-aligned models into testable hypotheses for Neuro?

---

> > > > > > ### Author Response · Authors · 2025-12-02
> > > > > > **Follow-up response to reviewer gx3G**
> > > > > >
> > > > > > ## Reviewer follow-up #3:
> > > > > >
> > > > > > `I like this idea a lot, and the use of representational geometries to probe it. How would one ultimately translate those differences between AT models vs. human-aligned models into testable hypotheses for Neuro?`
> > > > > >
> > > > > > ## Response:
> > > > > >
> > > > > > We thank the reviewer for their interest. We agree that the differences between AT models and human-aligned models are worth exploring, and that they may offer a path towards testable hypotheses about how the brain encodes the visual world. As noted earlier, prior work [1] has shown that AT models exhibit certain desirable manifold properties (compactness and separability), similar to neurally guided models. However, our newly added analyses (Appendix D.5, Fig. 15B) show that AT models do not converge towards the human-like representational geometry. This suggests that the key properties that make human representations unique may lie in the fine-scale, nonlinear geometric structure of human representational spaces that the current manifold statistics only partially capture. For example, one could predict that higher-order VVS regions contain nonlinear features such as curvature or local anisotropies that may be absent in AT models but can be inherited by NG models. Such properties could be measured in neural representational space using techniques that estimate local geometry (e.g., geodesic distances along neural manifolds). These measurements, in turn, could continue to generate concrete hypotheses about which geometric features contribute to adversarial robustness.
> > > > > >
> > > > > > We also note that understanding these finer-scale geometric properties would be valuable for neuroscience more broadly, as they offer a geometric viewpoint on how the brain encodes the visual world which is an idea that connects naturally with the recent trend in the neuroscience field that takes a manifold view for understanding neural population encoding [2].
> > > > > >
> > > > > > More broadly, regarding how we can “use DCNNs as an experimental platform for testing hypotheses”, the space of possibilities is indeed vast. For instance, one could test the hypothesis that feedback connections are necessary to achieve human-like representational organization by incorporating feedback mechanisms into DCNNs and examining the resulting representational geometry and downstream behavioral signatures. We agree that there are indeed many interesting directions to be explored in the future.
> > > > > >
> > > > > > [1] Dapello et al., 2021. Neural Population Geometry Reveals the Role of Stochasticity in Robust Perception. NeurIPS
> > > > > >
> > > > > > [2] Perich et al., 2025 A neural manifold view of the brain. Nature Neuroscience

---

> > > > ### Comment · Reviewer_gx3G · 2025-11-25
> > > > **Response**
> > > >
> > > > "Regarding why we use adversarial examples to probe category manifolds" I disagree with this logic. If the authors believe human data is needed to norm an OOD object recognition benchmark but none exists, then the authors should collect those data. Those results would ground the findings on adversarial robustness. I find the arguments about on/off manifold hollow when the most important question for neuroscience is whether you can transfer object constancy from human recordings to models.
> > > >
> > > > "Regarding why we broadly focus on adversarial robustness" What about evaluating how aligning to fMRI data affect tolerance to changes in object scale? That has been linked to pooling operations along the visual hierarchy, minimizing the need for mental rotation or attentional operations.

---

> > > > > ### Author Response · Authors · 2025-12-02
> > > > > **Follow-up response to reviewer gx3G**
> > > > >
> > > > > ## Reviewer follow-up #2:
> > > > >
> > > > > `"Regarding why we use adversarial examples to probe category manifolds" I disagree with this logic. If the authors believe ... "Regarding why we broadly focus on adversarial robustness" What about evaluating how aligning to fMRI data ...`
> > > > >
> > > > > ## Response:
> > > > >
> > > > > **OOD, including scaling, robustness is beyond the scope of the present work**: As we have discussed above, prior studies and our own preliminary results (not included) show that neurally-guided and bio-inspired models, while improved in adversarial robustness, do not show comparable outcomes on OOD transformations. This likely reflects current limitations in alignment methods which produces models with representations that only partially capture human neural representations. Better alignment will enable a fuller assessment on which human invariances could be transferred to DCNNs. We agree with the reviewer that, at that time, testing OOD robustness and collecting human behavioral as well as neural data would strengthen claims about VVS function. This is an interesting and important avenue to explore, but we leave this systematic examination of robustness to scale, pose, and other OOD transformations and its relation to adversarial attacks to future work and further model improvements.
> > > > >
> > > > > **OOD evaluation does not necessarily ground adversarial robustness**: Beyond the dissociations between them in neurally-aligned models as discussed above, this has also been observed for standard models [1]. Prior work [2] has even shown potential *trade-offs* between adversarial and certain spatial robustness types including rotation and translation. Thus, OOD results, critically, would not necessarily “ground” adversarial findings as the reviewer suggests. Instead, they may probe different mechanisms entirely.
> > > > >
> > > > > **On/off manifold consideration is critical rather than hollow**: The manifold hypothesis is one of the theoretical motivations and an important hypothesis we tested in the current work. That manifolds should contain identity-preserving transformations of objects is a critical assumption of this framework. We would like to make sure that we do not violate this assumption so that our tests are valid. Our goal in this work is not to determine exhaustively which transformations fall into that set, so we use adversarial perturbations because they, as almost imperceptible perturbations, naturally satisfy this requirement.
> > > > >
> > > > > **Regarding “the most important question for neuroscience is whether you can transfer object constancy from human recordings to models”**: As discussed in the response above, we agree that this is an important question for studying VVS function and should be explored in future work. At the same time, we think that understanding how the human visual system organizes representations is also a central question. Our results contribute to this by showing, for the first time, that biological representations evolve along the VVS towards progressively more compact and more linearly separable manifolds. This structure appears sufficient to confer hierarchical gains in adversarial robustness in models, thus revealing interesting geometrical properties of the representational space that biological systems possess and that standard DCNNs lack.
> > > > >
> > > > > We hope our clarifications above and in the revised manuscript (Introduction + Appendix F) now make clear that we are testing a specific hypothesis about the VVS representational space, the manifold geometry and adversarial robustness, not claiming to explain all aspects of the human robustness.
> > > > >
> > > > > [1] Taori et al., 2020. Measuring robustness to natural distribution shifts in image classification. NeurIPS
> > > > >
> > > > > [2] Kamath et al., 2021. Can we have it all? on the trade-off between spatial and adversarial robustness of neural networks. NeurIPS

---

> > > ### Comment · Reviewer_gx3G · 2025-11-25
> > > **Response**
> > >
> > > 3/4 Thank you.
> > >
> > > Questions:
> > >
> > > 1/2/3 Thank you.

---

> > ### Comment · Reviewer_gx3G · 2025-11-25
> > **Response**
> >
> > 1. I get your argument, but it feels like it's sidestepping the most obvious way that supervising on higher-order visual regions should help a DNN, which is to learn stable representations of objects over transformations. I understand that adversarial perturbations are an important line of research for DNNs, but object constancy is what has been studied for decades in neuroscience in the context of the visual hierarchy. I think the important missing experiment is to test DNN accuracy on objects with poses, lighting, scale, etc. that are out-of-distribution of ImageNet. There are lots of existing datasets designed to test this, for example ObjectNet. Otherwise, while I totally get your argument about the limitations of ImageNet, I do not understand how to square the findings that aligning to higher regions of the visual system would give more adversarial robustness but lower accuracy (which I would interpret as reflecting a loss in invariance), while lower regions would give the opposite pattern.
> >
> > This is my fundamental problem with the current work. If you can address it I will raise my score.
> >
> > 2. Thank you for this.

---

> > > ### Author Response · Authors · 2025-12-02
> > > **Follow-up response to reviewer gx3G**
> > >
> > > ## Reviewer follow-up #1:
> > >
> > > ` I get your argument, but it feels like it's sidestepping the most obvious way that supervising on higher-order visual regions should help a DNN, which is to learn stable representations... `
> > >
> > >
> > > ## Response:
> > >
> > > **Regarding the focus on adversarial versus OOD robustness**: We understand the reviewer’s desire for us to test our neurally guided models on the real-world invariances that humans exhibit. Indeed, understanding such invariances remains an important question in neuroscience and testing which invariances transfer through neural alignment is an interesting direction for future work, although **it is outside the scope of our present study**.
> > >
> > > As we clarify in the newly added Appendix F and as reviewer GQbY also notes, it has already been shown that while the neurally aligned or bio-inspired models show promise for adversarial robustness, they currently do not show comparable advantages for OOD transformations ([1] [2] ,CORnet models, as shown on the BrainScore leaderboard, and our own preliminary explorations (not included)). It remains an important question as to why this dissociation exists. One important possibility is that current alignment methods provide only a partial shift towards neural representational structure, as indicated by the small nudge by neurally guided models shown in Figure 2C. Consequently, we do **not** claim, or expect, that neural guidance leads to a full spectrum of human object recognition capabilities.
> > >
> > > Instead, we view our focused results in the current work as an important foundation for a broader investigation in the future. We here demonstrate, for the first time, the hierarchically evolving representational geometry across human VVS to support adversarial robustness and that manifold geometry is one potential mechanism. Going forward, stronger neural alignment could be pursued to allow comprehensively evaluating the gains on both adversarial robustness and OOD performance on benchmarks such as ObjectNet or ImageNet-A or -C. However, such a systematic investigation falls outside the scope of the present work.
> > >
> > > **Regarding the accuracy-robustness tradeoff**: We appreciate the opportunity to clarify this further. The tradeoff does **not** indicate loss of invariance, but rather should be viewed as a positive sign that the model is being steered away from brittle cues and towards more robust ones. This reflects a well-documented tension in supervised learning on natural datasets [3][4]. Naturalistic datasets (including but beyond ImageNet) contain many statistically predictive cues (textures, backgrounds) that humans rarely use to accurately classify objects. In standard training, the sole objective of maximizing classification accuracy forces models to exploit these cues to maximize their performance. Methods that improve robustness, on the other hand, including both adversarial training *and* neural guidance as shown in the current work, necessarily steer the models away from highly predictive shortcut cues that are not used by the human brain and that ultimately do not confer robustness. Therefore, this trade-off is about feature selection rather than lost invariance.
> > >
> > > [1] Safarani et al., 2021. Towards robust vision by multi-task learning on monkey visual cortex. NeurIPS 34.
> > >
> > > [2] Malik et al., 2023 Extreme image transformations affect humans and machines differently. Biological Cybernetics
> > >
> > > [3] Tsipras et al. 2018 Robustness may be at odds with accuracy. arXiv:1805.12152.
> > >
> > > [4] Ilyas et al. 2019 Adversarial Examples Are Not Bugs, They Are Features. NeurIPS, 32

---

### Official Review · Reviewer_zmbi · 2025-10-31

**Soundness:** 3
**Presentation:** 2
**Contribution:** 3
**Rating:** 6
**Confidence:** 3

**Summary:**

This paper is motivated by the robstness of the human visual system to adversarial examples, which have been studied extensively in the computer vision literature. To better understand this phenomenon, the VVS in human brains is modeled using deep neural networks. Using fMRI data from humans looking at images, neural predictors are traubed to model the responses of regions of interest in the brain to unseen images. These predictions are then used to align the outputs of neural heads, which are added to traditional ResNet image classifiers. A related training method where representations are aligned with compact manifolds is also defined in order to test the manifold disentanglement hypothesis. Experiments show that neural guidance does in fact increase robustness, and the hierarchy of robustness when aligning models with different regions of interest aligns with that of the human visual system. Further experiments explore the learned representation spaces of the NG-models and demonstrate the effectiveness of the manifold guidance method.

**Strengths:**

* The results in this paper are surprising : simply training a model to mimic human brain activity confers adversarial robustness. Not only could this provide insight into the robustness of the human visual system to image perturbations, it could also inform our knowledge of machine learning robustness more generally.
* The results provide compelling evidence for the manifold disentanglement hypothesis and may be of note to cognitive scientists.
* The novel manifold guidance loss term that is defined in this paper may inspire future research in training models that are robust to image corruptions.

**Weaknesses:**

* The model presented doesn't engage much with the existing literature on adversarial robustness. I think that additional space should be devoted to the relationship between this paper and the adversarial training literature in the related work. It would be informative to know whether adversarially trained models also conform to the manifold disentanglement hypothesis.
* The robustness results would be more compelling if they weren't restricted to $\ell_p$ bounded adversarial corruptions. There are many types of common (non-adversarial) corruptions that humans are robust to which we want our models to be invariant to (such as those included in Imagenet-C [1]). These corruptions can also be used to define adversarial bounds which are distinct from $\ell_p$ (i.e. [2]). It's not clear from the results presented that these models are robust to anything other than $\ell_p$ imperceptible corruptions.
* Relatedly, I think it's important to note that just because a model is more robust to a specific adversarial attack, it is not necessarily more robust to adversarial attacks in general (as noted in [3]). It is possible that an informed adversary could design an attack that effectively targets neurally-guided models. I think that this should be noted as a limitation.
* Experiments were only run using fMRI data from four individuals. This low sample size may be unavoidable due to difficulties in collecting data.

[1] Hendrycks, Dan, and Thomas Dietterich. "Benchmarking neural network robustness to common corruptions and perturbations." arXiv preprint arXiv:1903.12261 (2019).

[2] Kaufmann, Max, Daniel Kang, Yi Sun, Steven Basart, Xuwang Yin, Mantas Mazeika, Akul Arora et al. "Testing robustness against unforeseen adversaries." arXiv preprint arXiv:1908.08016 (2019).

[3] Tramer, Florian, Nicholas Carlini, Wieland Brendel, and Aleksander Madry. "On adaptive attacks to adversarial example defenses." Advances in neural information processing systems 33 (2020): 1633-1645.

**Questions:**

* How does the performance of a DNN trained with neural guidance compare to an adversarially trained model in terms of robustness to adversarial examples?
* Are the results architecture dependent? Did you run experiments on anything other than ResNet architectures?
* Is the manifold guidance method of training useful outside of the neural guidance framework? Would you expect to see similar improvements in robustness if the manifolds for each class were not derived from neural data and instead chosen in some other (possibly unsupervised) way?
* How do your neural and manifold guidance loss terms impact the time complexity of model training?

---

> ### Author Response · Authors · 2025-11-21
> **Responses to reviewer zmbi**
>
> We thank the reviewer for their valuable comments. Below are our point-to-point responses.
>
> ## Weaknesses
>
> ### 1. The model presented...
>
> As also discussed in General Response #2, we have added additional experiments to show that adversarial training (AT) presents as another engineering-driven approach that is effective, but works through mechanisms different from humans. We expand on this below.
>
> We would first like to note that our goal is not to develop a new defense method but to use DCNNs as an experimental platform for testing hypotheses about what supports human’s robustness. In this context, we view AT as another engineering-driven intervention, similar to our weight decay controls. Both of them improve robustness, but neither induces representational changes that resemble the human ventral stream.
>
> To illustrate this distinction, we have trained three adversarially trained models with increasing perturbation bounds ($\epsilon$= 0.003, 0.009, 0.015, see Appendix D.5, Fig. 15). As expected, AT produced substantial gains in robustness. However, importantly, these models showed representational geometries that remained very close to standard DCNNs and far from human VVS representations (Fig. 15B), paralleling weight decay models. Neural guidance, expectedly, consistently produced a shift towards human ventral stream geometry.
>
> Regarding manifold properties, prior work has shown that AT does improve manifold statistics [1]. As we have discussed in General Responses #3, this provides valuable supporting evidence for a link between adversarial robustness and manifold geometry. However, prior work did not test the core questions of the present study, i.e., whether similar manifold properties emerge across the human VVS, and whether imposing human-derived manifold constraints alone is sufficient to confer robustness. We have updated our related work section to discuss this.
>
> [1] Dapello et al., 2021 Neural Population Geometry Reveals the Role of Stochasticity in Robust Perception. NeurIPS, 34
>
> ### 2. The robustness results...
>
> We thank the reviewer for raising this important point. We agree that robustness to naturalistic and other OOD corruptions is an important capability for both biological and artificial visual systems. However, as also discussed in General Response #3, we here prioritized adversarial attacks because of the heterogeneous OOD findings in prior work and the specific theoretical needs for our manifold analysis.
>
> First, prior work on neural alignment has shown that neural alignment's benefits for broad OOD benchmarks are often heterogeneous. For instance, Safarani et al. [1] found that V1-aligned models improved robustness only against specific noise types. Interestingly, bio-inspired models like VOneNet also show performance degradation on certain OOD transformations [2]. Furthermore, our preliminary exploration with the popular ImageNet-C benchmark qualitatively mirrored this pattern: neurally guided models improved robustness, even hierarchically, to some corruptions but not to others.
>
> We argue that this heterogeneity highlights two critical implications that motivate our focus on adversarial robustness:
>
> * Less clear link to representational geometry: Such mixed findings suggest that “OOD robustness” is not a unitary phenomenon and may not have a straightforward link to representational geometry. Instead, it is a broad umbrella comprising diverse transformations that likely recruit distinct, specialized cognitive mechanisms. In contrast, adversarial robustness has shown a link to manifold geometry, as prior work [3] has demonstrated that adversarially trained models seem to develop more compressed and linearly separable manifolds.
>
> * Less guaranteed to be on-manifold: Our study tests the manifold disentangling theory where one essential claim is that category manifolds consist of identity-preserving transformations. However, humans are not fully foolproof for these OOD transformations, showing performance degradation just like the DCNNs on transformations like adding uniform noises or even changing contrast [4]. Since human performance on many OOD benchmarks remains uncharacterized, they are less ideal for strictly characterizing manifold geometry. On the other hand, the nature of almost imperceptibility for adversarial perturbations more safely ensures they mostly remain “on-manifold” for humans, making them the more precise choice given our focus on these manifold structures.
>
> We have added Appendix F to make this focus explicit.
>
> [1] Safarani et al., 2021 Towards robust vision by multi-task learning on monkey visual cortex. NeurIPS 34.
>
> [2] Malik et al.,  2023 Extreme image transformations affect humans and machines differently. Biological Cybernetics, 117(4), 331-343.
>
> [3] Dapello et al.,  2021 Neural Population Geometry Reveals the Role of Stochasticity in Robust Perception. NeurIPS 34
>
> [4] Geirhos et al., 2018 Generalisation in humans and deep neural networks. NeurIPS 31.

---

> > ### Author Response · Authors · 2025-11-21
> > **Responses to reviewer zmbi**
> >
> > ### 3. Relatedly,...
> >
> > Thank you for raising this important point. We agree that improved robustness to a specific attack does not guarantee robustness to all others, especially in the presence of an adaptive adversary. We note that our goal in this work is not to propose a new defense method that is universally robust. Rather, we use adversarial perturbations as an experimental probe to test whether and how ventral visual stream representations support human adversarial robustness. The replication of the improvement hierarchy across mainstream adversarial attack benchmarks shown in Appendix C.2 and Fig. 8 serves to confirm that our results are not coincidental to a specific kind of attack picked.
> >
> > Relatedly, similar to what the reviewer notes, humansmay not be fully foolproof as well. Recent studies [1][2] have shown that it is not entirely impossible to attack human perception with such adversarial manipulations, particularly when viewing time is limited and thus feedback and recurrent connections are less likely to engage. We have already discussed this in the paper, noting that the human visual system may not be fully immune to adversarial attacks and also that we do not claim that the neurally guided models have achieved human-level robustness.
> >
> > [1] Elsayed et al., 2018 Adversarial examples that fool both computer vision and time-limited humans. NeurIPS, 31.
> >
> > [2] Guo et al., 2022 Adversarially trained neural representations are already as robust as biological neural representations. ICML
> >
> > ### 4. Experiments were...
> >
> > Thank you for raising this concern. Indeed, limited sample size is one inherent limitation that comes fMRI. Collecting large scale neural data at this resolution and stimulus breadth similar to the NSD used here is even more challenging, which is why only a small number of participants exist in the publicly available dataset. However, we have tried to replicate the main finding on three additional NSD subjects who completed the full imaging sessions, and the hierarchical pattern was consistent despite variability in neural signal quality. We hope that this cross-subject replication offers some reassurance that the effects are not idiosyncratic to a single participant.
> >
> > ## Questions
> >
> > ### 1. How does the perf...
> >
> > We have discussed this point in detail in Weakness #1 above. To summarize here, we have added new experiments in Appendix D.5 and Fig.15 with models trained with AT. We first show that AT yields larger absolute gains in adversarial robustness, which is expected for a method explicitly optimized as a defense against adversarial attacks. However, as shown in Fig.15B, AT behaves similarly to weight decay in our analyses. It provides strong robustness improvements but does not shift model representations towards human VVS geometry. Neural guidance, in contrast, indeed produces representational changes that align more closely with human neural data. This suggests that the human representational space may offer a uniquely different pathway to robustness. We have added this discussion to both Appendix D.5 and main text section 4.2.
> >
> > ### 2. Are the results...
> >
> > We thank the reviewer for raising this important point. As also discussed in General Response #2, we have added additional experiments with VGG-16 and observed the same hierarchical robustness trend, thus showing that the effect is not tied to a specific architecture. The corresponding results are now included in Appendix C.5 and Fig.11.
> >
> > ### 3. Is the manifold...
> >
> > Thank you for raising this interesting question. Manifold guidance could indeed have the potential to be applied independently. One recent follow-up work by Chung and colleagues [1] demonstrated that a manifold-inspired objective (maximizing inter-manifold margins) can be implemented in a self-supervised setting. While promising for improving downstream task performance, their formulation similarly relies on linear approximations and therefore does not model non-linear properties. Furthermore, they treat category manifolds as fully independent, while psychological research on categorization [2] suggests that category structure is often highly correlated and hierarchical, thus leaving room for future extensions.
> >
> > In the present work, we chose to use manifold properties estimated directly from human ventral stream data because our primary goal was to test whether human manifold geometry relates to robustness. We agree, however, that exploring the applicability of manifold guidance is an exciting direction for future research. Such work may also further help clarify which manifold properties are needed to close the robustness gap observed here between manifold guidance and full neural guidance.
> >
> > [1] Yerxa et al., 2024 Contrastive-equivariant self-supervised learning improves alignment with primate visual area IT. NeurIPS 37
> >
> > [2] Rosch, E. (1978). Principles of categorization. In E. Rosch & B. Bloom (Eds.), Cognition and categorization (pp. 27–48). Routledge.

---

> > > ### Author Response · Authors · 2025-11-21
> > > **Responses to reviewer zmbi**
> > >
> > > ### 4. How do your neural and manifold guidance...
> > >
> > > For both neural guidance and manifold guidance, the additional time complexity during training is quite minimal relative to the vanilla image classification training setting.
> > >
> > > For neural guidance, all neural representations can be precomputed before training. During training, the only extra computation beyond the baseline is thus evaluating a small linear projection layer and computing the MSE loss for the neural head. If n denotes the dimensionality of the model’s final representation, this adds an overhead on the order of O(n^2) for each image, which is a negligible increment given the training cost of the baseline vanilla network.
> > >
> > > For manifold guidance, all manifold quantities such as the category manifold center, radii, and covariance matrices can likewise be precomputed and stored. During training, the extra operations are evaluating the same small linear projection and computing the manifold loss. This again produces an overhead on the order of O(n^2) for each image, which remains very small compared with the overall training cost for the vanilla DCNN.

---

### Official Review · Reviewer_GQbY · 2025-11-01

**Soundness:** 3
**Presentation:** 3
**Contribution:** 4
**Rating:** 6
**Confidence:** 5

**Summary:**

The paper investigates the neural mechanisms underlying human visual robustness and explores how these can be transferred to deep neural networks (DNNs). Specifically, the authors test the hypothesis that robustness in human vision arises progressively along the ventral visual stream (VVS)—from early visual areas (V1, V2) to higher-order regions (e.g., TO, PHC)—rather than being localized solely in specialized cortical areas.
To evaluate this, the authors train DNNs to align their internal representations with human neural responses from consecutive VVS regions using data from the Natural Scenes Dataset (NSD). They demonstrate that DNNs aligned to later visual areas exhibit greater adversarial robustness than those aligned to early areas, establishing a clear hierarchy. The study further connects this effect to the geometry of category manifolds in neural representational space—showing that smaller manifold extent and greater linear separability in higher-order regions predict improved robustness. Finally, the authors introduce “manifold guidance”, aligning DNN category manifold geometry (rather than individual activations) with human neural manifolds, and find that it qualitatively reproduces the hierarchical robustness pattern.

**Strengths:**

Strong experimental validation: multiple controls, cross-subject replication, and attack-type diversity.


Novel conceptual framing: robustness as a byproduct of manifold geometry inherited from neural data.


Elegant theoretical integration: combines neuroscience principles with modern robustness evaluation.


High reproducibility: clear methodological exposition, use of open datasets (NSD), and code availability.


Mechanistic insight: links representational geometry to robustness, moving beyond surface-level performance metrics.

**Weaknesses:**

The reliance on fMRI-based predictors limits representational precision; neural alignment fidelity could improve with higher-resolution data (e.g., intracranial recordings).


The scope of tasks (primarily object classification) is limited; extending analyses to temporal or contextual visual understanding would strengthen generalization claims.


Manifold guidance, while promising, remains a coarse approximation; incorporating nonlinear manifold constraints could further clarify its efficacy.


The current framework does not model feedback and recurrence, known to be crucial for human visual robustness.

**Questions:**

Could the authors quantify how fMRI signal noise or predictor quality influences the strength of the robustness hierarchy?


Would the same hierarchical effect persist if neural alignment were performed using multi-subject averaged representations rather than subject-specific ROIs?


Can the manifold guidance loss be extended to self-supervised or contrastive settings to assess scalability beyond supervised classification?


How do the authors interpret the trade-off between accuracy and robustness observed in later VVS-guided models?

The authors have a few missing citations: In line 40 around the unnatural image degradations, the authors should missed Extreme Image Transforms (EITs) [Crowder et al., 2022; Malik et al., 2023, Biol Cybernetics, Malik et al., 2023, arXiv] which present a novel view of structural changes in the input images.

When comparing previous work of Dapello et al., the authors should consider comparing their work to VOneNet [Dapello et al., 2020, NeurIPS], which fails under similar circumstances as EITs and is not widely tested on datasets like ImageNet-C [Hendrycks et al, 2019, ICLR].

The authors have uploaded a zip file for the code, but are encouraged to release it online on a publicly accessible platform.

---

> ### Comment · Reviewer_GQbY · 2025-11-20
>
> Please let me know if there are any questions about the review. Thanks.

---

> ### Author Response · Authors · 2025-11-21
> **Responses to reviewer GQbY**
>
> We thank the reviewer for their valuable comments. Below are our point-to-point responses.
>
> ## Weaknesses
>
> ### 1. The reliance on ...
>
> We thank the reviewer for raising this important point. We agree that fMRI comes with inherent limitations in signal quality [1] and is a more indirect measure of neural activity than methods like intracranial recordings. However, we would like to note that fMRI offers unique strengths that are essential for the present study. That is, it allows simultaneous access to multiple ventral stream regions within the same human participants at a fine-grained spatial scale. This property makes it possible to examine representational progression across the ventral stream within a unified experimental framework. Intracranial coverage in humans is quite limited and usually dictated by clinical needs, making such systematic, end-to-end comparisons less feasible.
>
> Furthermore, we relied on the NSD [2], a large-scale 7T fMRI dataset which provides data from participants viewing a large (close to 10k) and diverse set of natural images. This is a data scale critical for training DCNNs that is rarely achievable for clinical patients from which intracranial data are usually obtained.The high field strength, the extensive data collected and the advanced preprocessing pipeline used in this dataset provide one of the best currently available human vision neural dataset.
>
> That being said, we agree that the limitations of fMRI should not be overlooked. Noise inherent in the fMRI recordings, for example, may also contribute to the modest magnitude of the absolute robustness gains observed in our study. We have added a discussion for the use of fMRI data and its limitations in the Discussion section of the revised manuscript.
>
> [1] Logothetis, 2008 What we can do and what we cannot do with fMRI. Nature, 453(7197), 869-878.
>
> [2] Allen et al., 2022 A massive 7T fMRI dataset to bridge cognitive neuroscience and artificial intelligence. Nat. Neurosci., 25(1), 116-126.
>
> ### 2. The scope of ...
>
> We agree that if ventral stream representations indeed become progressively more robust and disentangled across the hierarchy, such benefits should extend beyond object classification and to other vision tasks that also operate on such visual representations. We focused on object classification in this study because it is widely used as a probe of visual ability in both DCNNs and human psychophysics (such as categorization tasks and naming tasks), making it a natural starting point for testing our hypothesis.
>
> Importantly, we would like to note that we have included one additional task, image captioning, in our paper (Appendix C.3, Fig. 9 right). The hierarchical pattern is preserved in this setting as well, as reflected by the progressive improvements in BLEU scores under adversarial perturbations. This provides preliminary evidence that the effect may generalize beyond classification. That being said, we agree that further experiments on more complex temporal or context-sensitive visual tasks such as video understanding would be valuable for fully assessing the scope of the effect, and we do view this as a promising direction for future work.
>
> ### 3. Manifold guidance ...
>
> We agree that our linear approximation is a coarse representation and likely discards nonlinear structure that could otherwise fill the performance gap between neural guidance and manifold guidance.
>
> As briefly discussed in the paper, our current method essentially approximates category manifolds as linear high-dimensional ellipsoids. However, a large body of neuroscience research has shown that population neural responses form complex and highly curved manifolds [1] [2]. Exploring these nonlinear properties in the visual domain would be an important next step towards designing more expressive manifold constraints for DCNNs. One such future direction could be, for example, enforcing geodesic distance preservation (similar to nonlinear dimensionality reduction techniques like UMAP), allowing the model to respect the nonlinear topology of human representational geometry, although the practicality and computational cost of such constraints would need careful consideration.
>
> [1] Gallego et al., 2017 Neural Manifolds for the Control of Movement. Neuron, 94(5), 978–984.
>
> [2] Stringer et al., 2019 High-dimensional geometry of population responses in visual cortex. Nature, 571(7765), 361–365.

---

> > ### Author Response · Authors · 2025-11-21
> > **Responses to reviewer GQbY**
> >
> > ### 4. The current framework...
> > We thank the reviewer for highlighting this important and interesting direction. As we have discussed in the paper, the human visual system is indeed characterized by extensive recurrent and feedback connectivity [1]. These connections even outnumber the purely feedforward ones [2], and it is therefore unlikely that ventral stream representations emerge through a strictly feedforward process in the way they do in standard DCNNs. This architectural difference, together with differences in developmental and training environments, likely contributes to the difficulty that conventional DCNNs have in autonomously developing human-like representations that support robustness.
> >
> > There have been several recent efforts to incorporate recurrence or biologically inspired feedback into artificial models. For example, work by Konkle and Alvarez [3] demonstrated that adding simple recurrent mechanisms can produce measurable improvements in adversarial robustness. More sophisticated approaches in transformer-based models, such as the method proposed by Shi and colleagues [4] which implements a mechanism analogous to analysis-by-synthesis, have reported similar benefits. These studies suggest that recurrent structure could indeed be supporting more human-like and thus more robust representations.
> >
> > However, we would like to note that our goal in this current work is different. We are not aiming to propose new architectures with better robustness, but instead, we aim to test whether properties of ventral stream representations themselves support visual robustness. We directly test this by training an artificial system, i.e., DCNN to perform visual tasks while steering its representational space to be maximally similar to that of a given brain region. The hierarchical pattern of robustness that emerges from aligning to successive VVS stages thus supports the hypothesis that representational structure evolves progressively across the ventral stream. That being said, we suspect that recurrent architectures could indeed make alignment to ventral stream representations easier, and testing whether recurrence is necessary for developing these representational properties is an exciting direction for future research. Such work could also help clarify whether recurrent structure contributes to the disentangling of category manifolds across the ventral stream that we investigate here.
> >
> > [1] Felleman et al., 1991 Distributed Hierarchical Processing in the Primate Cerebral Cortex. Cerebral Cortex, 1(1), 1–47.
> >
> > [2] Markov et al., 2014 Anatomy of hierarchy: Feedforward and feedback pathways in macaque visual cortex. Journal of Comparative Neurology, 522(1), 225–259.
> >
> > [3] Konkle et al., 2023 Cognitive Steering in Deep Neural Networks via Long-Range Modulatory Feedback Connections. NeurIPS 37
> >
> > [4] Shi et al., 2023 Top-down visual attention from analysis by synthesis. CVPR
> >
> > ## Questions
> >
> > ### 1. Could the authors...
> >
> > Thank you for raising this important alternative explanation to rule out, as it would imply that the observed hierarchy might simply reflect differences in predictor quality or fMRI signal strength across regions. To evaluate this, we tested whether the performance of the neural predictors correlates with the robustness improvements obtained through neural guidance. We computed the correlation between the predictor performance and the robustness AUC from each brain region. There was no significant relationship between the two (Spearman’s rho=0.40, p=0.29). We then repeated the analysis using predictor performance normalized by the noise ceiling of each region, estimated via split-half correlation with Spearman-Brown correction to take into account fMRI signal quality. Again, there was no significant correlation (Spearman’s rho=0.23, p=0.55). These results suggest that neural predictor accuracy, even when accounting for fMRI signal quality in each brain region using noise ceiling estimates, does not fully account for the hierarchical robustness pattern observed.
> >
> > We have included this important analysis and results in Appendix B. We have also noted this in the results section 4.1.

---

> > > ### Author Response · Authors · 2025-11-21
> > > **Responses to reviewer GQbY**
> > >
> > > ### 2. Would the same...
> > >
> > > We agree that it would be more ideal and powerful to leverage the aggregate data from multiple subjects. However, unfortunately, directly averaging neural representations across subjects is not effective. Anatomical and functional cortical topology has been known to show high variability across individuals [1], and the lack of a one-to-one correspondence in representational spaces means that simply averaging would not preserve meaningful structure. Hyperalignment methods [2] have been proposed to address this issue, but these methods can still introduce distortions, and our earlier attempt to aggregate across subjects using hyperalignment methods show that DCNNs struggle to make use of the resulting shared space. In addition, we have also tried the multi-head approach described in [3], where one DCNN is trained with multiple neural heads, each aligned to one subject. This setup could potentially help strengthen the shared backbone with the larger data, but our preliminary results did not produce additional gains in alignment quality or adversarial robustness, which reinforces the difficulty of combining individuals’ data with current tools.
> > >
> > > That being said, very recent progress points toward a promising path forward. Emerging work on large scale brain foundation models aims to build shared representational spaces that integrate neural data from many individuals [4]. These models directly tackle the alignment problem and may eventually allow future work to leverage the full Natural Scenes Dataset or beyond, rather than subject-specific subsets. This could substantially increase the effective dataset size, improve the fidelity of the neural signal, and better assess the full potential of the VVS representational space. We have incorporated this into the discussion section in the revised manuscript.
> > >
> > > [1] Cox et al., 2003 Functional magnetic resonance imaging (fMRI) “brain reading”: detecting and classifying distributed patterns of fMRI activity in human visual cortex. Neuroimage, 19(2), 261-270.
> > >
> > > [2] Haxby et al., 2020 Hyperalignment: Modeling shared information encoded in idiosyncratic cortical topographies. elife, 9, e56601.
> > >
> > > [3] Allen et al., 2022 A massive 7T fMRI dataset to bridge cognitive neuroscience and artificial intelligence. Nat neurosci, 25(1), 116-126.
> > >
> > > [4] Wang et al., 2025 Foundation model of neural activity predicts response to new stimulus types. Nature, 640(8058), 470-477.
> > >
> > > ### 3. Can the manifold...
> > >
> > > Thank you for raising this interesting potential. We note that the original authors of the MFTMA framework has proposed a follow-up self-supervised learning method [1] that encourages high capacity and large inter-manifold margins. While they are able to improve downstream classification performance, they did not evaluate robustness, and their method also relies on linear approximations of manifold structure by assuming a high-dimensional ellipsoid. Furthermore, they treat category manifolds as fully independent, while psychological research on categorization [2] suggests that category structure is often highly correlated and hierarchical rather than independent. We appreciate the reviewer for raising this possibility of extension because there are interesting opportunities for future work that could better incorporate nonlinear structure and inter-manifold relationships more directly.
> > >
> > > Meanwhile, we still would like to note that in the present work, we focus on the manifold properties from the human VVS and test whether they potentially lead to adversarial robustness. For that reason, we used manifold parameters computed directly from neural data rather than uncovering its potential as an independent learning objective.
> > >
> > > [1] Yerxa et al., 2024 Contrastive-equivariant self-supervised learning improves alignment with primate visual area IT. NeurIPS, 37, 96045-96070.
> > >
> > > [2] Rosch, E. (1978). Principles of categorization. In E. Rosch & B. Bloom (Eds.), Cognition and categorization (pp. 27–48). Routledge.

---

> > > > ### Author Response · Authors · 2025-11-21
> > > > **Responses to reviewer GQbY**
> > > >
> > > > ### 4. How do the authors...
> > > >
> > > > We would like to note that a tradeoff between standard accuracy and adversarial robustness has been widely observed in the literature [1-3]. This line of past work has shown that, under supervised classification, there is an inherent tension between optimizing for standard accuracy and robustness. In naturalistic datasets such as ImageNet, many cues are highly predictive, but at the same time not stable under perturbations (e.g. cars must have roads in the background to be cars). Models are trained to maximize accuracy and therefore tend to rely on non-robust features that happen to correlate strongly with the label. Robust training methods, including the popular adversarial training (AT) that multiple reviewers mentioned, reduce dependence on these brittle cues by introducing uncertainty around them. However, this necessarily sacrifices the “easy” accuracy gains these cues provide.
> > > >
> > > > Our results here are indeed consistent with these past interpretations, although unlike AT, alignment to higher-order VVS regions could be steering the model towards human-aligned and robust features (e.g., holistic shape of objects) and away from the non-robust statistical shortcuts that DCNNs typically rely on. Consequently, the model loses the accuracy boost from those brittle features while gaining robustness. We note that this pattern also aligns with prior biological alignment work [4], which observed a similar trade-off when aligning models to mice’s V1 representations.
> > > >
> > > > We have added the interpretation of this observation more explicitly in the main results in section 4.1 of the revised manuscript.
> > > >
> > > > [1] Yang et al., 2020 A closer look at accuracy vs. robustness. NeurIPS 33.
> > > >
> > > > [2] Tsipras et al., 2018 Robustness may be at odds with accuracy. arXiv:1805.12152.
> > > >
> > > > [3] Ilyas et al., 2019 Adversarial Examples Are Not Bugs, They Are Features. NeurIPS, 32.
> > > >
> > > > [4] Li et al., 2019 Learning from brains how to regularize machines. NeurIPS, 32.
> > > >
> > > > ### 5. The authors have...
> > > >
> > > > We thank the reviewer for bringing our attention to this valuable work. We agree that these works on Extreme Image Transforms (EITs) are indeed all interesting evidence showing the robustness gaps between humans and machines. We have added these citations in the introduction.
> > > >
> > > > ### 6. When comparing...
> > > >
> > > > Thank you for highlighting these important results, as they provide important context regarding our focus on adversarial attacks as opposed to OOD robustness as we noted in detail in General Responses #3.
> > > >
> > > > In particular, as the reviewer noted, prior work [1] has shown that VOneNet, despite its biological grounding, struggles with the OOD transformations included in the EITs. Combined with similar findings from other neural alignment studies [2] and our own preliminary exploration, these observations suggest that OOD robustness is not a unitary phenomenon. Instead, it likely encompasses a diverse set of challenges that recruit distinct, specialized cognitive mechanisms in humans (if they are solvable at all). Consequently, the relationship between broad OOD robustness and neural representational geometry and specifically whether such transformations remain on the identity-preserving manifold remains less clear.
> > > >
> > > > We have updated our manuscript (Appendix F) to note these important prior results.
> > > >
> > > > [1] Malik et al., 2023 Extreme image transformations affect humans and machines differently. Biological Cybernetics, 117(4), 331-343.
> > > >
> > > > [2] Safarani et al., 2021 Towards robust vision by multi-task learning on monkey visual cortex. NeurIPS 34
> > > >
> > > > ### 7. The authors have uploaded...
> > > >
> > > > Thank you for the suggestion. We will release the full code on a publicly accessible repository upon deanonymization to ensure full transparency and reproducibility.

---

> > > > > ### Comment · Reviewer_GQbY · 2025-11-27
> > > > >
> > > > > Thank you for the detailed rebuttal response. i look forward to seeing these in the main text/appendix of the paper for further understanding of the work to the end user.

---

### Official Review · Reviewer_tPLc · 2025-11-03

**Soundness:** 3
**Presentation:** 3
**Contribution:** 2
**Rating:** 6
**Confidence:** 4

**Summary:**

This paper studies whether aligning DNNs to human neural responses with specifically-designed techniques can improve adversarial robustness. The authors proposed a dual-head architecture to train ResNet models with “neural guidance” by simultaneously optimizing classification tasks and alignment to fMRI responses from seven VVS regions. The study shows that guidance from higher-order VVS regions leads to greater robustness against adversarial attacks. To analyze and explain this phenomenon, they apply statistical modeling, Mean-Field Theoretic Manifold Analysis, and show that human neural category manifolds become less diffuse and more linearly separable along the VVS hierarchy. Finally, as opposed to point-to-point guidance, the authors introduce “manifold guidance” to match coarse geometric properties and effectively reproduce the hierarchical robustness effect.

**Strengths:**

- **Clear motivation and novel systematic investigation.** The paper studies and improves modes’ adversarial robustness via the alignment between DNNs and human visual systems. It provides a systematic examination of how robustness evolves across multiple consecutive human VVS regions, which is different from prior work that focuses on isolated areas such as just V1 or IT. This provides valuable insights into the hierarchical nature of visual robustness.
- **Rigorous experimental design and statistical analysis.** I find this paper especially interesting because it goes beyond only demonstrating the phenomenon but analyze how it occurs with the MFTMA analysis and further claims the importance of geometric structure with the manifold guidance experiments. The evaluation includes multiple baseline conditions, various NSD subjects (Figure 7), adversarial attacks and configurations (Figure 8), and tasks (Figure 9).
- **Clear Presentation.** The paper is well-written with clear communication of the methodology and evaluation. Well-designed visualizations help make the concepts of the alignment between human visual system and DNNs accessible.

**Weaknesses:**

We thank the authors for submitting the paper to ICLR 2026! There are a few weaknesses listed below which I believe can make the paper better.
- **Limited absolute robustness gain.** While the hierarchical pattern is consistent, with results from different NSD subjects, the absolute robustness improvements of TO-guided models are small. They still show substantial vulnerability to adversarial perturbations which are imperceptible to humans. The authors already acknowledge this limitation but more further insights on future approaches which can scale to human-level robustness or if there is an existence of fundamental barriers would make the work more significant to the field.
- **Manifold guidance limitations.** While manifold guidance reproduces the hierarchical pattern, the absolute improvements are very limited (Figure 4B). The linear approximation of manifold geometry seem to be too coarse. It would be good to see what additional information beyond manifold structure contributes to adversarial robustness in the full neural guidance condition.
- **Architecture generalization and other related work.** The paper focuses on ResNet architectures, it is unclear whether the findings can generalize to widely-used architectures such as vision transformers, other CNN architectures, and biologically-inspired architectures such as CORnet. Also, many defense mechanisms seem related but not discussed in the paper such as biologically-inspired defenses beyond neural alignment, adversarial training which also improves manifold geometric, etc.

**Questions:**

- Does the manifold geometry cause robustness, or do both come from some other property of neural representations? Could you test this by techniques such as directly controlling manifold properties independent of neural guidance?
- What if you train a model to align to different VVS regions simultaneously? Would this provide the model with more information and help improve robustness?
- There are many other robustness measures which come from human-model misalignment, such as common corruptions and out-of-distribution scenarios. Would the findings generalize to those scenarios?

---

> ### Author Response · Authors · 2025-11-21
> **Response to reviewer tPLc**
>
> We thank the reviewers for their valuable comments. Below are our point-to-point responses.
>
> ## Weaknesses
> ### 1. Limited absolute robustness gain
>
> We appreciate the reviewer’s observation and we agree that the absolute robustness gains are smaller than what one might expect from actual human performance. We believe that the small magnitude could primarily result from the limitations inherent to the neural data we rely on. fMRI data are known to be coarse and indirect measurements of neural activity. The necessary intermediate step of training neural predictors likely further introduces additional signal loss. These factors constrain how much information remains after being transferred from the human brain to the model, and the nature of fMRI, in particular, may place a more fundamental ceiling on the downstream performance improvements.
>
> However, we view this work as one of the initial steps towards probing the potential of human representational geometry, rather than an attempt to reach the absolute limit of the approach. One potential future direction, for example, could consider overcoming the difficulty of combining data across subjects. There are significant differences in anatomical and functional organization across individuals’ brains that make alignment across subjects difficult, and existing hyperalignment techniques [1] still introduce distortions that DCNNs seem to struggle with, given our preliminary experiments. As a result, each neural predictor is trained on only one subject’s data, and the limited scale of training data further constrains the quality of neural predictors. Recently emerging “brain foundation models” [2] that learn shared representational spaces across large populations could offer a promising path forward. By leveraging the full scale of datasets like NSD, these foundation models could provide neural representation with higher fidelity, closing the gap between model and human robustness.
>
> We have expanded the Discussion to explicitly address these barriers and future directions
>
> [1] Haxby, J. V., Guntupalli, J. S., Nastase, S. A., & Feilong, M. (2020). Hyperalignment: Modeling shared information encoded in idiosyncratic cortical topographies. elife, 9, e56601.
>
> [2] Wang, E. Y., Fahey, P. G., Ding, Z., Papadopoulos, S., Ponder, K., Weis, M. A., ... & Tolias, A. S. (2025). Foundation model of neural activity predicts response to new stimulus types. Nature, 640(8058), 470-477.
>
> ### 2. Manifold guidance limitations
>
> We agree with the reviewer that the absolute robustness gain from manifold guidance is smaller than that observed with full neural guidance, and that further work is necessary to determine what else might contribute to the robustness of the neurally-guided models.
>
> As discussed in the paper, one likely reason is the linear approximation we adopted. Our current method essentially approximates category manifolds as linear high-dimensional ellipsoids. However, a large body of neuroscience research has shown that population neural responses form complex and highly curved manifolds [1] [2]. Exploring these nonlinear properties in the visual domain would be an important next step towards designing more expressive manifold constraints for DCNNs. One such future direction could be, for example, enforcing geodesic distance preservation (similar to nonlinear dimensionality reduction techniques like ISOMAP or UMAP), allowing the model to respect the nonlinear topology of human representational geometry, although the practicality and computational cost of such constraints would need careful consideration.
>
> Beyond geometry, neural guidance should be transferring specific and critical feature biases leveraged by humans to achieve robustness. For instance, prior work [3] suggests that neural alignment steers models towards low spatial frequencies, which contains global shape information, and away from the high-frequency textures that DCNNs typically rely on. Such shift in feature prioritization need not be exclusively or fully explained by manifold geometry alone. However, we would like to note that we do not claim that manifold geometry is the only mechanism, but to test this hypothesis as a potential explanation and show that the human VVS representations, the disentanglement of category manifolds, and downstream adversarial robustness are linked. More comprehensive experiments will be needed to more definitively identify the mechanisms through which the human VVS supports robustness.
>
> [1] Gallego et al., 2017 Neural Manifolds for the Control of Movement. Neuron, 94(5), 978–984.
>
> [2] Stringer et al., 2019 High-dimensional geometry of population responses in visual cortex. Nature, 571(7765), 361–365.
>
> [3] Li et al., 2023 Robust deep learning object recognition models rely on low frequency information in natural images. PLOS Computational Biology, 19(3), e1010932.

---

> > ### Author Response · Authors · 2025-11-21
> > **Response to reviewer tPLc**
> >
> > ### 3. Architecture generalization ...
> >
> > We thank the reviewer for these insightful suggestions and address them below:
> >
> > Regarding architectural generalization: As also discussed in General responses #2, we trained an additional set of models using VGG-16 (Appendix C.5, Fig. 11). We observed the same hierarchical robustness progression as in the ResNet-based experiments. This confirms that the effect is not specific to architectural features (e.g., skip connections) of ResNets.
> >
> > Regarding biologically inspired architectures such as CORnet: We would like to note that they introduce a significant confound for our goal and experimental design. Because CORnet layers are explicitly mapped to brain regions (e.g., CORnet-V1 should function like human V1), aligning different brain regions to the same readout layer (as we do to control for depth) would violate the architecture’s design principles. Conversely, aligning brain regions to their corresponding CORnet layers would force us to vary the depth of the neural head simultaneously with the source of neural activity, making it harder to isolate the effect of brain regions from the architectural depth. We therefore chose generic backbones (ResNet, VGG) to provide a clean and controlled platform where only the source of neural guidance varies.
> >
> > Regarding other defenses such as adversarial training: As also discussed in General Response #2c, we would like to emphasize that our goal is not to propose a new defense method, but to use neural guidance to probe which aspects of human representational structure relate to robustness. However, we have added experiments (Appendix D.5) comparing Neural Guidance (NG) to Adversarial Training (AT). While AT yields high robustness, our representational similarity analysis (RSA) results show that AT models do not develop a human-like representational geometry. This highlights that NG and AT achieve robustness via fundamentally different mechanisms. Furthermore, we thank the reviewer for raising the prior work that shows AT also improves manifold statistics [1], as this provides valuable supporting evidence for a relationship between manifold geometry and adversarial robustness. However, prior work did not test the core questions of the present study: whether similar manifold properties emerge across the human ventral stream, and whether imposing human derived manifold constraints alone is sufficient to confer robustness. We have updated our Related Work to explicitly discuss this.
> >
> > [1] Dapello et al., 2021 Neural Population Geometry Reveals the Role of Stochasticity in Robust Perception. NeurIPS, 34, 15595–15607.
> >
> > ## Questions
> >
> > ### 1. Does the manifold geometry ...
> >
> > We would like to note that we view our manifold guidance setup as a more direct intervention to examine what neural representation properties could be the cause of robustness. In particular, although the manifold parameters were estimated from neural data, the model is trained only on these derived geometric quantities and never sees neural responses directly, thus effectively decoupling the geometric structure from the precise neural content. The resulting qualitative replication of the robustness gain hierarchy thus provides a more causal contribution of manifold geometry than prior correlation-based studies.
> >
> > However, we emphasize that our experiments do not claim that manifold geometry is the sole and definitive cause. Indeed, as the reviewer also raised in the previous section, there still remains a quantitative performance gap between manifold guidance and full neural guidance. It is possible, for example, that the manifold structure is a byproduct of other computational objectives in the ventral stream, such as building more global and holistic object shape representations [1].
> >
> > Regarding the possibility of direct manipulation of manifold properties independent of neural data, We note that one recent self-supervised learning method [2] have explored this by explicitly maximizing manifold capacity (large inter-manifold margins). While promising for improving downstream classification performance, robustness remains largely untested in that framework and their method similarly relies on linear approximations of manifold structure by assuming ellipsoids. At the same time, they treat category manifolds as fully independent, while psychological research on categorization [3] suggests that category structure is often highly correlated and hierarchical. We appreciate the reviewer for raising this possibility of extension because there are interesting opportunities for future work that could better incorporate nonlinear structure and inter-manifold relationships more directly.
> >
> > [1] Biederman 1987 Recognition-by-components: a theory of human image understanding. Psych. Rev., 94(2), 115.
> >
> > [2] Yerxa et al., 2024 Contrastive-equivariant self-supervised learning improves alignment with primate visual area IT. NeurIPS, 37, 96045-96070.

---

> > > ### Author Response · Authors · 2025-11-21
> > > **Response to reviewer tPLc**
> > >
> > > ### 2. What if you train ...
> > >
> > > If the reviewer suggests aligning the DCNN’s final representation to all regions simultaneously to leverage combined information, we hypothesize this may be less effective. Higher-order regions (like TO) likely already subsume the computational advantages of earlier ones. Furthermore, forcing the final layer to learn such diverse information across many brain regions could introduce conflicting constraints that confuses the model, thus potentially leading to worse alignment and diluting the robustness gains.
> > >
> > > However, if the reviewer refers to a hierarchical alignment (e.g., mapping V1 to early layers and TO to later layers), this could potentially be more effective. If the VVS builds representations in a scaffolded manner, this approach would help the DCNN construct a more faithful early foundation (e.g., V1-like representations), thereby facilitating better alignment with the higher-order regions that further support downstream robustness. However, such hierarchical alignment raises the challenge of determining which model layers should correspond to which brain regions. As deep networks and the ventral stream do not share a fixed one-to-one mapping, introducing multiple alignment targets at once requires a principled way to decide how information should flow across layers.
> > >
> > > We do notice a concurrent work, HOT, proposes learning a “transport plan” to align two arbitrary networks [1]. Exploring how to better align multiple brain regions simultaneously, and how such alignment might influence robustness, is an exciting direction for future research.
> > >
> > > [1] Shah et al., 2025 Representational Alignment Across Model Layers and Brain Regions with Hierarchical Optimal Transport. arXiv preprint arXiv:2510.01706.
> > >
> > > ### 3. There are many other ...
> > >
> > > As also detailed in General Response #3, we prioritized adversarial perturbations over broad OOD benchmarks (such as ImageNet-C) because of the heterogeneous OOD findings in prior work and the specific theoretical needs for our manifold analysis.
> > >
> > > First, prior work on neural alignment has shown that while alignment confers adversarial robustness, its benefits for broad OOD benchmarks are often heterogeneous. For instance, Safarani et al. [1] found that V1-aligned models improved robustness only against specific noise types, while sometimes performing worse on others. Interestingly, as noted by Reviewer GQbY, bio-inspired models like VOneNet also show performance degradation on certain OOD transformations [2]. Furthermore, our preliminary exploration with the popular ImageNet-C benchmark qualitatively mirrored this pattern: neurally guided models improved robustness, and even hierarchical robustness, to some corruptions but not to others.
> > >
> > > We argue that this heterogeneity highlights two critical implications that motivate our focus on adversarial robustness:
> > >
> > > * Less clear link to representational geometry: Such mixed findings suggest that “OOD robustness” is not a unitary phenomenon and may not have a straightforward link to representational geometry. Instead, it is a broad umbrella comprising diverse transformations that likely recruit distinct, specialized cognitive mechanisms. In contrast, adversarial robustness has shown a link to manifold geometry, as prior work [3] has demonstrated that adversarially trained models seem to develop more compressed and linearly separable manifolds.
> > > * Less guaranteed to be on-manifold: Our study tests the manifold disentangling theory where one essential claim is that category manifolds consist of identity-preserving transformations. However, humans are not fully foolproof for these OOD transformations, showing performance degradation just like the DCNNs on transformations like adding uniform noises or even changing contrast [4]. Since human performance on many OOD benchmarks remains uncharacterized, they are less ideal for strictly characterizing manifold geometry. On the other hand, the nature of almost imperceptibility for adversarial perturbations more safely ensures they mostly remain “on-manifold” for humans, making them the more precise choice given our focus on these manifold structures.
> > >
> > > We have added Appendix F to make this reasoning explicit.
> > >
> > > [1] Safarani et al., 2021 Towards robust vision by multi-task learning on monkey visual cortex. NeurIPS 34.
> > >
> > > [2] Malik et al., 2023 Extreme image transformations affect humans and machines differently. Biological Cybernetics, 117(4), 331-343.
> > >
> > > [3] Dapello et al., 2021 Neural Population Geometry Reveals the Role of Stochasticity in Robust Perception. NeurIPS, 34, 15595–15607.
> > >
> > > [4] Geirhos et al., 2018 Generalisation in humans and deep neural networks. NeurIPS 31.

---

### Author Response · Authors · 2025-11-21
**General Responses**

We thank all reviewers for the thoughtful and constructive feedback. Given the common themes across reviewers, we would like to clarify our scientific goal, summarize 3 additional experiments  (VGG-16, random seeds, and comparisons to Adversarial Training (AT)), and clarify our focus on adversarial attacks rather than broad out-of-distribution (OOD) benchmarks.
1. **Scientific goal**: We would like to clarify that our primary goal is not to propose a new SOTA defense method. Rather, we aim to use deep convolutional neural networks (DCNNs) as an experimental platform for examining whether and how representational geometry across the ventral visual stream (VVS) supports humans’ visual robustness. While we hope the principles revealed in this work could inspire future bio-inspired defenses, our contribution is the mechanistic insight linking VVS representational geometry to robustness, rather than performance gains over engineering methods.

2. **New experiments**: We have added three sets of results to address concerns regarding generalizability and comparison to AT:

* Architecture generalization (Appendix C.5, Fig.11): To address architecture dependence (reviewers 8mca, tPLc, zmbi), we replicated our main results using VGG-16. We observed the same hierarchical progression, i.e., guidance from later VVS regions yields greater robustness, confirming that the results are not idiosyncratic to ResNet. Note: per reviewer 8mca, we have updated “DNNs” to “DCNNs” throughout the manuscript to more precisely reflect our scope.

* Statistical reliability (Appendix C.4, Fig.10): To ensure our results are robust to random seed initialization (reviewer 8mca), we trained a subset of models across 6 additional random seeds to show that robustness curves are highly stable and reliable.

* Comparison to adversarial training (Appendix D.5, Fig. 15): To distinguish our method from engineering approaches (reviewers zmbi, gx3G), we trained 3 additional models using AT with varying perturbation bounds. While AT, expectedly, increases robustness, we found that AT models do not align with human representational geometry, and instead cluster closely with vanilla models in representational similarity space, thus reaffirming that neural guidance confers robustness via a distinct and biologically grounded mechanism compared to engineering approaches.

3. **The focus on adversarial attacks instead of OOD benchmarks**: Reviewers (tPLc, GQbY, Zmbi, gx3G) asked to consider broader OOD benchmarks and why we prioritized adversarial attacks instead. We clarify below and also in the new Appendix F (also referenced in the introduction) that this is driven by both the mixed findings in prior work and our specific theoretical focus of probing manifold geometry.

    First, prior work has shown heterogeneous benefits for broad OOD benchmarks from neural alignment. For instance, Safarani et al. [1] found that V1-aligned models improved robustness only against a subset of ImageNet-C noise types, and Malik et al. [2] found that other bio-inspired models like VOneNet also perform worse on certain OOD transformations. Our preliminary exploration with ImageNet-C also qualitatively mirrored this pattern.

    We emphasize that this heterogeneity highlights two critical implications that motivate our specific focus on adversarial robustness:

    * Less clear link to representational geometry: Such heterogeneity suggests that “OOD robustness” is not a unitary phenomenon with a clear link to representational geometry. Instead, it is a broad umbrella comprising diverse transformations that likely recruit distinct, specialized cognitive mechanisms. In contrast, prior work [3] has shown that adversarially robustified models develop more compressed and linearly separable manifolds.

    * Less guaranteed to be “on-manifold”: Our study tests the manifold disentangling hypothesis, where one essential claim is that category manifolds consist of identity-preserving transformations. However, human performance on many OOD benchmarks remains uncharacterized and certain OOD transformations such as adding uniform noise or even changing contrast [4] could also confuse humans. In contrast, the almost imperceptible nature of adversarial perturbations is more likely to ensure that they remain “on-manifold” (identity-preserving) for humans, making them more natural choices given our focus on manifold structures.

We respond to each reviewer point-by-point below. All updates in the manuscript are highlighted.

[1] Safarani et al., 2021 Towards robust vision by multi-task learning on monkey visual cortex. NeurIPS 34.

[2] Malik et al., 2023 Extreme image transformations affect humans and machines differently. Biological Cybernetics, 117(4), 331-343.

[3] Dapello et al., 2021 Neural Population Geometry Reveals the Role of Stochasticity in Robust Perception. NeurIPS 34.

[4] Geirhos et al., 2018 Generalisation in humans and deep neural networks. NeurIPS 31.

---

### Meta-Review · Area_Chair_pDEw · 2026-01-05

**Summary:**

This paper proposes a neurally guided training setup motivated by human/brain robustness and evaluates whether it yields more robust DNNs under adversarial-style perturbations. Reviewers found the experiments careful, but several felt the contribution is incremental and that the paper mostly (and not fully convincingly) ends up describing another way to improve adversarial robustness, without clearly establishing any novel insight about human vision. The AC agrees with these main criticisms.

**Reviewer Concerns:**

Main concerns raised by reviewers, which remain central:
- limited novelty relative to prior work connecting robustness, adversarial training, and neural alignment
- unclear/limited contribution beyond demonstrating improved adversarial robustness
- narrow focus on adversarial perturbations as a proxy for human robustness, with little connection to broader notions of robustness (image degradations, invariances, etc)
- unclear advantage over simpler baselines such as standard adversarial training, and limited mechanistic insight into what the neurally guided component adds

The AC feels that the rebuttal does a good job of providing details, adding experiments, additional architectures, and clarifications of scope. It improves presentation and empirical coverage, but largely reframes or defends the approach rather than resolving the core concerns. In particular, the rebuttal does not convincingly establish stronger novelty, does not broaden the notion of robustness being tested, and does not clarify a clear scientific takeaway about human vision. As a result, the AC feels that the main criticisms stand.

**Reviewer Scores:**

Pre-rebuttal scores: one clearly positive review and several borderline-accept reviews, with one clear reject. Large shifts seem unlikely.
* 8mca: 8, likely unchanged (clearly positive)
* GQbY: 6, likely unchanged
* tPLc: 6, likely unchanged or small increase
* zmbi: 6, likely unchanged or small increase
* gx3G: 2, likely unchanged (core objections remain)

---

### Decision · Program_Chairs · 2026-01-26

Reject